



**1**  **Impacts of Global Wildfire Aerosols on Direct Radiative, Cloud and**

**2**  **Surface-Albedo Effects Simulated with CAM5**

Yiquan Jiang[1,2], Zheng Lu[2], Xiaohong Liu[2,*], Yun Qian[3], Kai Zhang[3], Yuhang Wang[4]
and Xiu-Qun Yang[1]
[1] *CMA-NJU Joint Laboratory for Climate Prediction Studies, Institute for Climate*
*and Global Change Research, School of Atmospheric Sciences, Nanjing University,*
*Nanjing, China*
[2] *Department of Atmospheric Science, University of Wyoming*
[3] *Pacific Northwest National Laboratory, Richland, Washington, USA*
[4] *School of Earth and Atmospheric Sciences, Georgia Institute of Technology, Atlanta,*
*Georgia, USA*
*Corresponding author:
Dr. Xiaohong Liu
Department of Atmospheric Science
University of Wyoming
Laramie, WY 82071
Phone (307) 766-3225
E-mail: xliu6@uwyo.edu





**Abstract**
Aerosols from wild-land fires could significantly perturb the global radiation
balance and induce the climate change. In this study, the Community Atmospheric
Model version 5 (CAM5) with prescribed daily fire aerosol emissions is used to
investigate the spatial and seasonal characteristics of radiative effects (REs) of
wildfire aerosols including black carbon (BC) and particulate organic matter (POM).
The global annual mean direct radiative effect (DRE) of all fire aerosols is $0.155 \pm$
$0.01$ W m$^{-2}$, mainly due to the absorption of fire BC ($0.25 \pm 0.01$ W m$^{-2}$), while fire
POM induces a small overall effect (-0.05 to $0.04 \pm 0.01$ W m$^{-2}$). Strong positive DRE
is found in the Arctic and in the oceanic regions west of South Africa and South
America as a result of amplified absorption of fire BC above low-level clouds, in
general agreement with satellite observations. The global annual mean cloud radiative
effects (CRE) due to all fire aerosols is $-0.70 \pm 0.05$ W m$^{-2}$, resulting mainly from the
fire POM indirect effect ($-0.59 \pm 0.03$ W m$^{-2}$). The large cloud liquid water path over
land areas of the Arctic favors the strong fire aerosol indirect effect (up to -15 W m$^{-2}$)
during the Arctic summer. Significant surface cooling, precipitation reduction and
low-level cloud amount increase are also found in the Arctic summer as a result of the
fire aerosol indirect effect. The global annual mean surface albedo effect (SAE) over
land areas ($0.03 \pm 0.10$ W m$^{-2}$) is mainly due to the fire BC-in-snow effect (0.02 W
m$^{-2}$) with the maximum albedo effect occurring in spring (0.12 W m$^{-2}$) when snow
starts to melt.





## 1. Introduction

Wildfires or biomass burning of living and dead vegetation are an integral component of the Earth system, and have significant impacts on the carbon cycle [*Ciais et al.*, 2013] and the climate [*Bowman et al.*, 2009; *Keywood et al.*, 2011; *Liu et al.*, 2014; *Sommers et al.*, 2014]. On one hand, wildfires can perturb the climate system by emitting greenhouse gases and aerosols [*Kaiser et al.*, 2012; *Wiedinmyer et al.*, 2011]. On the other hand, climate states and variabilities can play a critical role in determining the occurrence frequency and intensity of wildfires [*Marlon et al.*, 2009; *van der Werf et al.*, 2008; *Westerling et al.*, 2006]. However, there are still large unknowns regarding the feedback mechanisms between wildfire and climate interactions, and more investigations are needed in order to predict the future wildfire events and their climatic impacts [*Carslaw et al.*, 2010; *Liu et al.*, 2014]

Particles emitted from wildfires can exert significant perturbations to the climate system by scattering and absorbing the solar radiation in the atmosphere (i.e., direct effect) [*Carslaw et al.*, 2010] and by changing the surface albedo when they are deposited on the snow and ice (i.e., surface albedo effect) [*Flanner et al.*, 2007; *Quinn et al.*, 2008; *Randerson et al.*, 2006; *Qian et al.*, 2011, 2015]. In addition, wildfire or smoke particles can modify the cloud properties, precipitation efficiency, and the hydrological cycle by changing the atmospheric thermal structure (i.e., semi-direct effect) [*Koch and Del Genio*, 2010; *Andreae et al.*, 2004b] or acting as cloud condensation nuclei (CCN) (i.e., indirect effect) [*Andreae and Rosenfeld*, 2008; *Qian et al.*, 2009; *Lu and Sokolik*, 2013].



The radiative effect (RE) [*Boucher and Tanre*, 2000] and radiative forcing (RF)
[*Forster et al.*, 2007; *Myhre et al.*, 2013a] are typical metrics used to assess and
compare anthropogenic and natural drivers of the climate change. RE represents the
instantaneous radiative impact of all atmospheric particles from both anthropogenic
and natural sources [*Heald et al.*, 2014]. RF is calculated as the change of RE from
pre-industrial (e.g., year 1850) to present-day (e.g., year 2000) [*Heald et al.*, 2014;
*Liu et al.*, 2007], based on the aerosol and precursor gas emissions in the
pre-industrial and present-day times [*Dentener et al.*, 2006; *Lamarque et al.*, 2010].
The direct radiative forcing (DRF) or forcing due to
Aerosol-Radiation-Interaction (ARI) of biomass burning aerosols has been estimated
since the IPCC second Assessment Report (AR2). In the IPCC second and third
Assessment Reports (AR2 and AR3), the DRF of biomass burning aerosols is about
-0.40 W m$^{-2}$ (ranging from -0.60 to -0.07 W m$^{-2}$) [*Houghton*, 1996; *McCarthy*, 2001].
In the IPCC Fourth Assessment Report (AR4), it is estimated to be about 0.03 W m$^{-2}$
(ranging from -0.09 to 0.15 W m$^{-2}$) [*Forster et al.*, 2007]. The more positive forcing
in the IPCC AR4 results from the improved representation of absorption properties of
biomass burning aerosols and the consideration of effects of low-level clouds on the
absorption of above-cloud biomass burning aerosols in the global models. Based on
the Aerosol Comparisons between Observations and Models (AeroCom) Phase II
simulations [*Bond et al.*, 2013; *Myhre et al.*, 2013b], the DRF of biomass burning
aerosols in the IPCC Fifth Assessment Report (AR5) is estimated to be 0.0 W m$^2$
(ranging from -0.20 to 0.20 W m$^{-2}$), and the DRFs of biomass burning black carbon





(BC) and primary organic matter (POM) are of the opposite sign (i.e., 0.10 and -0.10
W m$^{-2}$, respectively).

There are also many studies that estimated the direct radiative effect (DRE) of

biomass burning aerosols by comparing the simulation with fire emissions against the
simulation with no fire emissions. For example, using the NCAR Community
Atmosphere Model version 4 (CAM4) with a bulk aerosol module, *Tosca et al.* [2013]
reported that the top-of-atmosphere (TOA) DRE from global biomass burning
aerosols is 0.18 ± 0.10 W m$^{-2}$ averaged for the period of 1997-2009. *Ward et al.* [2012]
estimated the DRE from biomass burning aerosols in the pre-industrial (for the year
1850), present-day (for the year 2000), and future time periods (for the year 2100),
and found that the biomass burning aerosol DRE for the year 2000 is 0.13 W m$^{-2}$ and
-0.27 W m$^{-2}$ in all-sky and clear-sky conditions, respectively.

The cloud radiative effect (CRE) or effect due to Aerosol-Cloud-Interaction (ACI)

of biomass burning aerosols can be comparable to or even larger than the DRE [*Liu et*
*al.*, 2014]. The CRE of biomass burning aerosols was reported to be -1.16 W m$^{-2}$ for
the present day in *Chuang et al.* [2002]. With a global aerosol-climate model, the
CRE of biomass burning aerosols was estimated to range from -1.74 to -1.00 W m$^{-2}$
for the year 2000 in *Ward et al.* [2012]. The semi-direct radiative effect of biomass
burning aerosols is not independently assessed in IPCC reports. The magnitude was
reported to be about 7.0 W m$^{-2}$ in the Southern American biomass burning regions by
examining the radiative flux difference with and without the biomass burning aerosol
effect on clouds [*Liu*, 2005].



The radiative forcing or effect of BC from wildfires and other sources on snow
and ice has been estimated in previous studies. For biomass burning emissions with a
strong (1998) and weak (2001) boreal fire year, RE of fire BC-in-snow was estimated
to be 0.011 and 0.006 W m$^{-2}$, respectively [*Flanner et al.*, 2007]. *Randerson et al.*
[2006] reported that BC from a boreal forest fire deposited on snow and sea ice
introduced a global annual mean RE of 8±5 W per m$^2$ of burned area in the first year
when the fire happened. A summary of BC-in-snow forcing/effect can be found in
*Bond et al*. [2013]. They reported that the present-day RE of fire BC-in-snow ranges
from 0.006 to 0.02 W m$^{-2}$ based on previous studies [*Jacobson*, 2004; *Rypdal et al.*,
2009; *Skeie et al.*, 2011; *Hansen et al.*, 2005; *Flanner et al.*, 2007, 2009; *Koch et al.*,
2009].
Biomass burning aerosols can have significant impacts on global and regional
precipitation and atmospheric circulation. With the change of fire emissions from year
1860 to 2000, *Jones et al.* [2007] found that biomass burning aerosols decrease the
global near-surface air temperature by about 0.25°C, when considering the feedbacks
of sea surface temperature (SST) in the model. As shown in *Tosca et al.* [2013], the
direct and semi-direct effects of biomass burning aerosols reduce the precipitation
near the equator and weaken the Hadley circulation. With a regional climate model,
*Zhang et al.* [2009] found that biomass burning aerosols may warm and stabilize the
lower troposphere and thus re-enforce the dry season rainfall pattern in the Southern
Amazonia. The absorption of shortwave radiation by biomass burning BC could
increase the vertical stratification and inhibit both the cloud formation and



precipitation [*Ackerman et al.*, 2000; *Tosca et al.*, 2014]. In contrast, biomass burning
aerosols could invigorate the convective clouds [*Andreae et al.*, 2004a; *Koren et al.*,
2005] through suppressing warm rain processes in the convection, and enhance the
latent heat release at higher levels [*Andreae and Rosenfeld*, 2008].
In this study, we estimate the present day (from year 2003 to 2011) wildfire
(biomass burning) aerosol REs (DRE, CRE and SAE) using the NCAR Community
Atmosphere Model version 5.3 (CAM5) with the four-mode version of the modal
aerosol module (MAM4). We use two methods to calculate the DRE of biomass
burning aerosols (total, BC-only, and POM-only). The spatial and seasonal
characteristics of biomass burning aerosol REs, and the impacts on the global
precipitation and temperature are discussed. Compared to earlier studies of biomass
burning aerosol REs [*Tosca et al.*, 2013; *Ward et al.*, 2012], a number of
improvements are made in this study, which include (1) a higher model horizontal
resolution at 0.9° by 1.25° versus 1.9° by 2.5°, (2) the latest CAM5 model with
MAM4, (3) daily instead of monthly fire emissions, and (4) a new methodology to
more accurately diagnose the REs of biomass burning aerosols under the cloudy-sky
condition.
The paper is organized as follows. Section 2 introduces the model and
experiments. Section 3 describes the methods to diagnose the biomass burning aerosol
REs. Section 4 presents the model results of biomass burning aerosol REs, and
impacts on global and regional surface temperature and precipitation. Conclusions
and discussion are given in Section 5.




### 2.    Model, Experiment Design and Aerosol Radiative Effect Method

2.1 Model

In our study, we use the Community Earth System Model (CESM) version 1.2,

with the Community Atmosphere Model version 5.3 (CAM5.3) [*Neale et al.*, 2010]
coupled with the Community Land Model version 4 (CLM4) [*Oleson et al.*, 2010].
The SNow, ICe, and Aerosol Radiative model (SNICAR) [*Flanner and Zender*, 2005]
is turned on in the simulations to diagnose the biomass burning BC-in-snow effect.
CAM5 includes several major updates in its physics parameterizations compared to
previous CAM versions. A two-moment stratiform cloud microphysics scheme is
included in CAM5 to predict both the mass and number mixing ratios of cloud liquid
and cloud ice [*Morrison and Gettelman*, 2008]. MAM4, which was updated from the
three-mode version of the Modal Aerosol Model (MAM3) [*Liu et al.*, 2012], includes
aerosol mass and number mixing ratios in four lognormal modes: Aitken,
accumulation, coarse, and primary carbon mode [*Liu et al.*, 2016]. The primary
carbon mode is included to improve the treatment of microphysical ageing of BC and
POM, compared to MAM3. MAM4 significantly increases (and improves) the
near-surface BC concentrations in the Arctic with only a slight increase (~10%) in the
computational time [*Liu et al.*, 2016].

2.2 Experiment design

CAM5 was run with the finite volume dynamics core in a resolution of 0.9°

latitude by 1.25° longitude and 30 vertical levels. The model was run for the time





period of year 2003 to 2011 (i.e., for 9 years) with prescribed monthly sea surface
temperatures and sea ice. The year 2003 was run twice and the first year simulation
was used as a model spin-up. Global Fire Emissions Database version 3.1 (GFED 3.1)
daily emissions [*Giglio et al.*, 2013] for BC, POM and sulfur dioxide ($SO_2$) from 2003
to 2011 are prescribed, and the vertical distribution of fire emissions is based on the
AeroCom protocol [*Dentener et al.*, 2006]. Anthropogenic aerosol and precursor gas
emissions are from the IPCC AR5 dataset [*Lamarque et al.*, 2010]. We performed our
control experiment (FIRE) with the GFED fire emissions turned on and a sensitivity
experiment (NOFIRE) with the fire emissions turned off. Differences between FIRE
and NOFIRE experiments are used to calculate the REs and climate effects of
biomass burning aerosols. Two additional experiments (NOFIREBC and
NOFIREPOM) were performed with fire BC and POM emissions turned off,
respectively. Differences between the control (FIRE) and these two experiments
represent the contribution from biomass burning BC and POM, respectively. Other
forcings (e.g., SST, greenhouse gases) of all these experiments are kept the same. We
performed ten ensemble members for each of these experiments. Furthermore, we
performed the other experiment (FIRE_BBFFBF) using the modified CAM5 model
that separately predicts the BC and POM from biomass burning (BB), fossil fuel (FF)
and biofuel (BF) sources, while other model features are kept the same as the FIRE
experiment. A summary of all the experiments in this study can be found in Table 1.

2.3 Methods of calculating fire aerosol radiative effects





The REs of all fire aerosols, fire BC, and fire POM are calculated from the
differences of TOA shortwave fluxes ($\Delta F$) between the FIRE experiment and the
three other experiments (NOFIRE, NOFIREBC and NOFIREPOM), respectively.
$$\Delta F_{fire\ aero} = F_{fire} - F_{nofire} \tag{1}$$
$$\Delta F_{fire\ bc} = F_{fire} - F_{nofirebc} \tag{2}$$
$$\Delta F_{fire\ pom} = F_{fire} - F_{nofirepom} \tag{3}$$
The total TOA shortwave flux change can be broken into the aerosol direct
radiative effect (DRE, i.e., radiative effect from aerosol-radiation interactions), the
aerosol induced cloud radiative effect (CRE, i.e., radiative effect from aerosol-cloud
interactions), and the surface albedo effect (SAE, i.e., radiative effect from
aerosol-surface albedo interactions). The aerosol induced CRE results from both the
aerosol indirect effect on clouds via acting as CCN and the aerosol semi-direct effect
on clouds via affecting the atmospheric states due to absorbing aerosols. We adopt the
method of *Ghan* [2013] to separate the DRE, CRE, and SAE from the total effects of
all fire aerosols, fire BC and fire POM, respectively. The method is summarized as
follows. $F_{clean}$ is the radiative flux at TOA calculated from a diagnostic radiation call
in the same control simulations, but neglecting the scattering and absorption of solar
radiation by aerosols. $F_{clean,clear}$ is the clear-sky radiative flux at TOA calculated from
the same diagnostic radiation call, but neglecting scattering and absorption by both
clouds and aerosols.
$$\Delta F = \quad \Delta(F - F_{clean}) \quad + \quad \Delta(F_{clean} - F_{clean,clear}) \quad + \quad \Delta F_{clean,clear} \tag{4}$$
$\quad\quad\quad\quad$ (DRE) $\quad\quad\quad\quad\quad\quad$ (CRE) $\quad\quad\quad\quad\quad\quad$ (SAE)





In the method above, CRE includes both aerosol indirect and semi-direct effects.
The fire BC has a much weaker indirect effect due to its lower mass burden and lower
hygroscopicity compared to fire POM [*Koch et al.*, 2011]. Thus the fire aerosol
semi-direct effect can be approximately estimated by the CRE of fire BC. The fire
aerosol indirect effect can be estimated as the difference of fire aerosol CRE and
semi-direct effect.
We undertake another method to estimate the fire aerosol DRE from the
experiment (FIRE_BBFFBF). With explicit predictions of fire POM and fire BC in
FIRE_BBFFBF, the DREs of fire BC and fire POM are estimated by two diagnostic
radiation calls, neglecting the scattering and absorption of solar radiation of fire BC
and fire POM, respectively. This method is named as BBFFBF, and the DREs of fire
BC and fire POM will be compared with those from the method of *Ghan* [2013]. The
fire BC-in-snow effect is calculated from SNICAR, and compared with the SAE
estimated from *Ghan* [2013].

**3. Results**

3.1 Simulation of biomass burning aerosols
The biomass burning BC and POM from forest and grass fires are significant
contributors to the total BC and POM emissions. Figure 1 shows the seasonal
variation of GFED fire emissions in the global, tropical (25°S to 25°N), and Arctic
(60°N to 90°N) regions. Global fire emission is the largest during the boreal summer
as well as in the boreal autumn (September/October), when it is the fire season in the



tropical regions of the Southern Hemisphere (SH). The tropical fire emission
contributes the most to the annual global fire emission (80% for BC and 85% for OC,
respectively), compared to other regions. Arctic is the other important fire region,
where the emission maximum is found during the summer. In summer, the OC
emission in the Arctic regions is about 50% of that in the tropical region. The BC
emission in the Arctic is much smaller than that of the tropical regions even in the
summer fire season. The dominant fire type in the SH topics is deforestation, savanna
and grassland fires, while that in the Arctic is the forest fires. The OC to BC ratio
(OC/BC) of forest fires is almost three times higher than that of deforestation, savanna
and grassland fires [*van der Werf et al.*, 2010].
Figure 2 shows the latitudinal and longitudinal distributions of vertically
integrated concentrations (column burdens) of BC and POM from BB, FF, and BF
sources based on the FIRE_BBFFBF experiment. The BC and POM from BB source
are mainly distributed in the tropical and sub-tropical regions (South Africa, South
America and Southeast Asia) and in the mid- to high latitudes (North of 45°N) of the
Northern Hemisphere (NH) (Northeast Asia, Alaska and Canada). The largest column
burdens of biomass burning aerosols are located in South Africa and adjacent oceanic
areas (1.5 and 20 mg m$^{-2}$ for BC and POM, respectively). The biomass burning
aerosols are important aerosol species in the Arctic regions, and contribute up to 53%
and 86% to the total burden of BC and POM in the Arctic (from 60° N to 90°N),
respectively. In comparison, the maximum column burdens of fossil fuel BC and
POM are found in East Asia, South Asia, Western Europe and North America. The



maximum column burdens of biofuel BC and POM occur in East Asia, South Asia
and Central Africa. The biofuel and fossil fuel sources are dominant contributors to
BC and POM in East Asia and South Asia. In other regions of the world, biomass
burning is the primary source of BC and POM. Globally, the biomass burning
contributes 41% and 70% to the total burdens of BC and POM, respectively. Biomass
burning can also emit $SO_2$. However, it only contributes ~3% to the total global
sulfate burden (figure not shown), so only radiative effects of biomass burning POM
and BC are discussed in this study.

The simulated aerosol optical depth (AOD) and single scattering albedo (SSA)

are validated with observations from the AErosol RObotic NETwork (AERONET,
http://aeronet.gsfc.nasa.gov) at sites significantly affected by biomass burning
activities in South Africa, South America and the Arctic regions. The AERONET
AOD and SSA data are averaged for the years from 2003 to 2011 to match the
simulation period. We note that *Tosca et al.* [2013] and *Ward et al.* [2012] applied
scaling factors (from 1 to 3 varying by regions) to fire emissions to improve modeled
AOD magnitudes. In South Africa, modeled monthly AOD agrees with observations
within a factor of 2 for the three sites (Figure 3a-3c). The underestimation of AOD is
found in the tropical site (Mongu) (Figure 3a) during autumn (the fire season). The
simulated AOD in the two other sites (Skukuza and Ascension Island) is generally
consistent with observations in both the magnitude and seasonal trend. The simulated
SSA in South Africa ranges between 0.75 and 0.95 and generally matches the
observed SSA magnitude and trend in the two land sites (Mongu and Skukuza)





(Figure 4a-4b). However, an overestimation of SSA is found in the oceanic site
(Ascension Island) (Figure 4c). The reason for this overestimation of SSA and thus
the underestimation of absorption AOD (AAOD) is unclear and could be due to that
the model has not treated the absorption enhancement of aged fire BC during its
transport.

The simulated AOD in South America is generally consistent with observations

within a factor of 2 (Figure 3d-3f). The seasonal variation of simulated AOD
generally matches the observations. The underestimation of AOD in Alta Floresta and
Cuiaba-Miranda is most obvious in September and October (the fire season), which
may be attributed to the underestimation of fire emissions. The simulated SSA in
South America ranges mostly between 0.87–0.95 and matches the observations
reasonably well (Figure 4d-4f).

In the Arctic, small AOD (less than 0.3) and large SSA (larger than 0.9) are

observed for the three sites. The large SSA in the fire season (summer) is consistent
with the high OC/BC ratio of fire emissions in the Arctic (Figure 1). The model
significantly underestimates the observed AOD in the Arctic in both fire and non-fire
seasons. The underestimation of AOD can be due to (1) the underestimation of fire
emissions in the NH high latitudes [e.g., *Stohl et al.*, 2013] and/or fossil fuel
emissions in Asia [e.g., *Cohen and Wang*, 2014], (2) the excessive scavenging of
aerosols during their transport from the NH mid-latitude industrial regions by
liquid-phase clouds [*Wang et al.*, 2013], and (3) the coarse horizontal resolution
(~100 km) of the model [*Ma et al.*, 2014]. Although MAM4 increases the column



burdens of POM and BC by up to 40 % in many remote regions compared to MAM3,
it still underestimates the surface BC concentrations in the Arctic [*Liu et al.*, 2016].
The modeled SSA in the Arctic is lower than observations, which implies that the
simulation of AAOD is better than that of AOD and the underestimation of
non-absorbing aerosols (e.g., sulfate) in the Arctic may be more severe than that of
BC.

3.2 Direct radiative effect

The annual mean DREs of all fire aerosols (including BC, POM and sulfate), fire

BC and POM, estimated with the method of BBFFBF and with the method of *Ghan*
[2013] are shown in Figure 5, respectively. The fire sulfate is not included in the
calculation of DRE of all fire aerosols with the method of BBFFBF. Its effect is minor
since the global annual mean burden of fire sulfate (0.09 mg m$^{-2}$) is much smaller than
that of fire POM (1.25 mg m$^{-2}$), both of which are light-scattering. The DRE of all fire
aerosols from the two methods agree with each other very well. The global annual
mean DRE of all fire aerosols is positive (0.155$\pm$0.01 W m$^{-2}$), which indicates a
warming effect from all fire aerosols. The DRE is positive on the globe except in
some land areas (e.g., South Africa, South America, Great Lakes, North Canada, and
East Siberia). The maximum positive DRE is located in ocean areas west of South
Africa (~5.0 W m$^{-2}$) and South America (~1.5 W m$^{-2}$). The positive DRE up to 1 W
m$^{-2}$ is found in the Arctic (60°N to 90°N). The different signs of DRE between land
and ocean areas of South Africa and South America result from the differences in


cloud fraction and cloud liquid water path (LWP) between land and ocean regions. In
the fire season (August-September-October) of the tropical regions, cloud fraction and
cloud LWP over the land areas (10% and 20 g m$^{-2}$, respectively) are much smaller
than those over the adjacent ocean areas (70% and 70 g m$^{-2}$, respectively). The
biomass burning aerosols are transported above the low-level stratocumulus clouds,
and their absorption is amplified by these clouds [*Abel et al.*, 2005]. A comparison of
modeled DRE in autumn (September-October-November) over the South Atlantic
Ocean with satellite observations is shown in Figure 6. The observed above-cloud
aerosol DRE is calculated with the method of *Zhang et al.* [2014] using the
Aqua/MODIS and Terra/MODIS products, respectively. The observed above-cloud
aerosol DRE over southeastern Atlantic Ocean is 3-12 W m$^{-2}$, with higher values near
the coasts. The simulated DRE agrees better with Terra/MODIS observed DRE than
with Aqua/MODIS in both the magnitude and spatial pattern.

The seasonal variation of DRE of all fire aerosols is shown in Figure 7. The DRE

has a maximum (1.13 W m$^{-2}$) in the boreal summer (June-July-August, JJA) over the
NH high latitudes. The maximum positive DRE in the tropical regions occurs in the
summer and autumn (September, October and November, SON) during the fire
season of South Africa and South America. The DRE reaches a positive maximum in
Southeast Asia during the fire season in March, April and May (MAM).

The DRE of fire BC is shown in Figure 5c-5d. The fire BC DRE calculated from

the two methods are similar in magnitudes and spatial patterns, and there are much
less noises from the BBFFBF method. The global annual mean fire BC DRE is about





$0.25 \pm 0.01$ W m$^{-2}$ and positive over the globe (the regions with negative values in
Figure 5d are in general not statistically significant). Unlike all fire aerosols, fire BC
generates a positive forcing in the land regions of South Africa and South America,
and the amplification effect of low-level clouds on fire BC positive forcing can be
clearly seen in South Africa and adjacent Atlantic Ocean.

The global annual mean DRE of fire POM from the two methods somewhat

differs from each other (Figure 5e-5f). The BBFFBF method gives a small negative
value (-0.05 W m$^{-2}$), while the *Ghan* [2013] method shows a small positive value
($0.04 \pm 0.01$ W m$^{-2}$). The difference is mainly in the Arctic regions where the positive
forcing from *Ghan* [2013] is larger than that from the BBFFBF method. This is
because the removal of fire POM emissions in the NOFIREPOM experiment affects
the burden of co-emitted fire BC, causing the decrease of BC burden in the Arctic (by
~0.05 mg m$^{-2}$) compared to the FIRE experiment. Thus, it should be careful in using
the *Ghan* [2013] method to diagnose the radiative forcing of a single component
within co-emitted aerosols. The DRE of fire POM is negative in most of the global
regions. However, positive forcing can be found over oceanic regions west of South
Africa and South America, North Pacific Ocean and the Polar regions where large
amount of low-level clouds, sea ice or land ice exist. The multiple scatterings between
the above-cloud fire POM and low-level clouds or between the fire POM and the
Earth's bright surface with high albedos could reduce the amount of solar radiation
reflected by these low-level clouds and bright surface in the case without the fire
POM. With the BBFFBF method the sum of DRE from fire POM and fire BC (i.e.,



0.20 W m$^{-2}$) is larger than that of all fire aerosols (0.15 W m$^{-2}$). It reflects the
nonlinear interactions among different aerosol components [*Ghan* et al., 2012]. The
nonlinearity is stronger with the *Ghan* [2013] method. The reason is that removing the
emission of one species (e.g., fire POM in the NOFIREPOM experiment) can affect
the burden of other co-emitted species (e.g., fire BC).

3.3 Cloud radiative effect
The annual mean CREs due to all fire aerosols, fire BC, and fire POM are shown
in Figure 8. The CRE diagnosed with the *Ghan* [2013] method includes both aerosol
indirect and semi-direct effects. The fire aerosol semi-direct effect (to be discussed
below) is much smaller (-0.04±0.03 W m$^{-2}$ on the global mean) than the indirect
effect, and the CRE is mostly from the fire aerosol indirect effect. The global annual
mean CRE of all fire aerosols is -0.70±0.05 W m$^{-2}$. In the tropical regions, the strong
negative CRE is located in the adjacent ocean areas of South Africa, South America
and Australia, with the maximum CRE of -8.0 W m$^{-2}$ over the South Atlantic Ocean.
The strong negative fire aerosol CRE also occurs in the Arctic (60°N to 90°N). The
CRE in East Siberia, Alaska and Canada is as large as -6.0 W m$^{-2}$.
The fire BC has a weak indirect effect by acting as CCN, but can reduce the cloud
amount through its semi-direct effect. The CRE of fire BC (Figure 8b) can
approximate the fire BC semi-direct effect with a small global annual mean value of
-0.04±0.03 W m$^{-2}$. However, stronger positive effect can be found in the western
Pacific (3.0 W m$^{-2}$) and Arctic regions (1.0 W m$^{-2}$). The global annual mean CRE of



fire POM is $-0.59 \pm 0.03$ W m$^{-2}$ (Figure 8c), and dominates the cloud effect of all fire
aerosols. The sum of CRE from fire BC and POM ($-0.62 \pm 0.03$ W m$^{-2}$) is smaller
than that of all fire aerosols ($-0.70 \pm 0.05$ W m$^{-2}$) due to the non-linear interactions of
fire BC and fire POM as well as the negative CRE of fire sulfate.

The seasonal variation of all fire aerosol CRE is shown in Figure 9. The

maximum of fire aerosol CRE is in the boreal summer (i.e., the fire season in NH)
located in the NH high latitudes (60°N to 90°N). The largest summer CRE is found in
the land areas and is as large as $-15$ W m$^{-2}$. The fire aerosol CRE in the tropical
regions is most significant in the boreal summer (up to $-15$ W m$^{-2}$) and autumn (up to
$-10$ W m$^{-2}$) over the ocean areas. The different spatial distributions of fire aerosol
CRE in the NH high latitudes and in the tropics result from the difference in cloud
distributions between the two regions. During the fire season the cloud LWP over the
land areas in the NH mid- and high latitudes is three times larger than that over the
ocean areas in the tropics. Larger cloud LWP favors the stronger CRE. Like the fire
aerosol DRE, the smallest fire aerosol CRE occurs in the boreal spring.

Seasonal variations of zonal mean fire aerosol DRE, CRE, cloud LWP, low-level

cloud amount, and vertically-integrated (burden) concentrations of fire POM and fire
BC are shown in Figure 10. The seasonal variation of fire BC and fire POM burdens
is largest in the SH low latitudes (from 30°S to 0°N) and NH mid- and high latitudes
(50°N to 90°N). Distinct features of these two areas can also be noticed that the
maximum fire BC burden in NH (0.3 mg m$^{-2}$) is much lower than that in SH (0.8 mg
m$^{-2}$), while the maximum POM burdens in these two areas are comparable.





Interestingly, the DRE is larger in the NH summer than that in the SH autumn
although the fire BC burden is much lower in the NH summer. It is mainly due to the
larger amount of low clouds in the NH high latitudes, which enhances the absorption
of fire BC. The maximum DRE in the NH summer is found near the North Pole
(70 °N to 90 °N), and not around 60 °N where the fire aerosol burden is highest. The
CRE of fire aerosols is about 3 times larger in the NH summer than that in the SH
autumn, although the burden of fire POM in NH is comparable to that in SH. The
larger cloud LWP in the NH summer around 40-60°N and higher fire OC/BC ratios
favor the stronger CRE there.

3.4 Surface snow albedo effect

Here we compare the modeled BC-in-snow (BCS) concentrations with

observation data collected from multiple field campaigns over the Arctic [*Doherty et*
*al.*, 2010] and Northern China [*Wang et al.*, 2013; *Qian et al.*, 2014]. Figure 11a
shows the simulated (from FIRE and NOFIRE experiments) and observed BCS
concentrations as a function of latitude. The range of observed BCS concentrations is
between 1 and 200 ng g$^{-1}$ in the Arctic and between 50 and 2000 ng g$^{-1}$ in Northern
China, respectively. Both FIRE and NOFIRE experiments capture the meridional
gradient in BCS concentrations between the mid-latitudes (Northern China) and high
latitudes (Arctic). The mean and median concentrations of BCS are both
overestimated in Northern China, implying the high biases from the anthropogenic
emissions and/or model physics (Figure 11b). The mean and median BCS





concentrations from the FIRE experiment agree better with observations than those
from the NOFIRE experiment in the Arctic (Figure 11b). This suggests that fire
emissions are important for BCS concentrations in the Arctic.

The annual mean SAE of all fire aerosols diagnosed from *Ghan* [2013] and the

fire BCS effect diagnosed from SNICAR are shown in Figure 12. The global annual
mean SAE ($0.03 \pm 0.10$ W m$^{-2}$) is much smaller compared to the DRE and CRE. The
SAE over land is maximum in spring ($0.12 \pm 0.27$ W m$^{-2}$) and winter ($0.06 \pm 0.16$ W
m$^{-2}$). The SAE over land in summer and autumn is very small (less than 0.01 W m$^{-2}$).
We note that the mean SAE calculated with *Ghan* [2013] is much smaller than the
standard deviation resulted from the internal variability.

The annual mean fire BCS effect calculated from SNICAR is shown in Figure

12b and 12c. The spatial distribution of the fire BCS effect is similar to the fire SAE,
implying that the fire SAE has a significant contribution from the fire BCS effect.
Averaged when only snow is present, the fire BCS effect is larger (0.048 W m$^{-2}$). The
global mean fire BCS effect (with the presence of snow) can be as large as 0.06 W m$^{-2}$
in spring, and the maximum effect (up to 1 W m$^{-2}$) is located in the Arctic regions
(East Siberia, Alaska and Greenland, figure not shown). The positive SAE in Siberia,
North America and Canada can be a result of BCS effect. However, the SAE in these
regions is larger than the BCS forcing especially in spring. The snow melting and
snow depth change due to the BCS warming may induce a larger positive SAE than
the albedo change due to BCS itself. The negative SAE over land is a result of the
snow depth change caused by fire aerosols.




3.5 Fire aerosol effects on shortwave radiation, global temperature and precipitation

Here, we show the annual mean net shortwave flux change at TOA (i.e., total

radiative effect), in the atmosphere and at surface, and changes in surface air
temperature, convective and large-scale precipitation due to all fire aerosols in Figure
13 and Table 2. The global mean net shortwave flux change at TOA due to all fire
aerosols is -0.55±0.07 W m$^{-2}$, which indicates that fire aerosols lead to the reduction
of shortwave flux into the Earth's system. The zonal mean TOA shortwave flux
reduction in the Arctic regions (-1.35±1.03 W m$^{-2}$) is much larger than that in the
tropical regions (-0.66±0.09 W m$^{-2}$). The cooling at TOA is mostly from fire aerosol
CRE. The maximum negative RE is located in the land areas of the Arctic and ocean
areas of the tropics. Although the global mean total radiative effect is negative,
positive effect is found in some land areas (e.g., Africa, Greenland).

The shortwave flux change in the atmosphere of the tropical regions is much

larger than that of the Arctic regions. It is because BC burden in the tropics (0.17 mg
m$^{-2}$) is larger than that in the Arctic (0.09 mg m$^{-2}$). Strong absorption (~8 W m$^{-2}$) in
the atmosphere is found in the land areas of South Africa and South America and in
the Southeast Atlantic. The surface shortwave flux change in the Arctic is mostly
from the TOA shortwave flux reduction due to the fire aerosol CRE, while the surface
shortwave flux change in the tropics is mostly due to the fire BC absorption in the
atmosphere.

The fire aerosols lead to the reduction of the global mean surface air temperature



($T_s$) by $0.03 \pm 0.03$ K, consistent with the reduction of shortwave fluxes at TOA and at
surface. The largest surface cooling is found in the Arctic and tropical regions by up
to 0.6 K. The cooling of the Arctic is related to the strong fire aerosol CRE, while the
cooling in the tropics is mainly from the surface shortwave flux reduction due to the
fire BC absorption. The $T_s$ change in the ocean areas is very small since the SST is
prescribed in our simulations.

The global mean total precipitation is reduced by $0.010 \pm 0.002$ mm day$^{-1}$ due to

all fire aerosols (Table 2). Unlike the $T_s$ change, the precipitation reduction in the
tropics (0.016 mm day$^{-1}$) is much larger than that in the Arctic (0.001 mm day$^{-1}$). The
reduction in the tropics is mainly from the large-scale precipitation decrease (0.015
mm day$^{-1}$). The net change in the convective precipitation is very small in the tropics
(0.001 mm day$^{-1}$), as the convective precipitation is significantly decreased near the
equator and increased in the regions away from the equator, consistent with the results
of *Tosca et al.* [2013]. The shortwave flux reduction at surface leads to a stabilization
of the atmospheric boundary layer and a suppression of the convection near the
equator. The strong atmospheric absorption by fire BC leads to the reduction of
low-level clouds and large-scale precipitation in the tropics. Both effects lead to a
significant reduction of total precipitation near the equator. The precipitation decrease
in the NH high latitudes is mainly from the reduction of convective precipitation.

Figure 14 shows the changes of $T_s$, total precipitation, cloud LWP, and low-level

cloud cover in the summer due to all fire aerosols. The $T_s$ is reduced by more than 1 K
in most of land areas around 60°N. The maximum cooling (larger than 1.5 K) is found





in East Siberia, Alaska and Canada. A decrease of total precipitation (by about 0.2
mm day$^{-1}$) is found in these regions. Accompanying the surface cooling and
precipitation reduction, a significant increase of cloud LWP and low-level cloud cover
is found there. This is a result of the indirect effect of fire aerosols in the land areas of
the Arctic (60°N to 90°N). The fire POM leads to the reduction of cloud droplet effect
radius and the increase of cloud droplet number concentration, consistent with
observed fire effects on clouds in Canada and the United States [*Peng et al.*, 2002].

**4. Discussion and Conclusions**
In our study, the fire aerosol radiative effect (RE) is calculated with CESM. The
method from *Ghan* [2013] is used to diagnose the DRE, CRE and SAE of fire
aerosols. Additional experiment with CESM which tracks the wildfire BC and POM
separately from fossil fuel and biofuel sources is performed to diagnose the fire
aerosol DRE and fire BC-in-snow effect for comparisons with the *Ghan* [2013]
method.
The BC and POM burdens from wildfires are largest in the tropical regions
(South Africa, South America and Southeast Asia) and in the NH mid- to high
latitudes (North of 45°N) (Northeast Asia, Alaska and Canada). Fire aerosols
contribute 41% and 70% to the global burden of BC and POM, respectively. When
comparing with the AERONET AOD and SSA data, modeled monthly AOD agrees
with observations within a factor of 2 for most of the South African and South
American sites. The model underestimation of AOD is found in the South American




sites near fire source regions, which is most obvious in the fire season (September and
October). The model underestimates the observed AOD in the Arctic regions in both
fire and non-fire seasons. The modeled SSA in South Africa and South America is
generally in agreement with observations, while the modeled SSA in the Arctic is
lower.

The annual mean DRE of all fire aerosols is $0.155 \pm 0.01$ W m$^{-2}$ and positive over

most areas except in some land areas (e.g., South Africa, North Canada, and East
Siberia). The annual maximum DRE is found in the oceanic areas west of South
Africa (5 W m$^{-2}$) and South America (1.5 W m$^{-2}$). The positive DRE over the land
regions of South Africa and South America is smaller, although the fire aerosol
burdens are higher. It is because the larger amount of low-level clouds in the oceanic
areas reflects the solar radiation back to the space for more absorption by fire BC
above clouds, and thus generates a larger positive DRE at TOA. The annual mean
DRE of fire BC is about $0.25 \pm 0.01$ W m$^{-2}$ and positive over the globe. Fire POM
induces a weak negative DRE globally (-0.05 W m$^{-2}$) with the BBFFBF method and a
small positive value ($0.04 \pm 0.01$ W m$^{-2}$) with the *Ghan* [2013] method. The positive
DRE of fire POM is found over oceanic areas west of South Africa and South
America, North Pacific, and polar regions where the low-level cloud coverage is large
or the surface albedo is higher. The maximum DRE in the Arctic regions occurs in the
summer (0.35 W m$^{-2}$), while the DRE in the tropical regions reaches its maximum in
the autumn.

The global annual mean CRE of all fire aerosols is $-0.70 \pm 0.05$ W m$^{-2}$ and the





maximum forcing is located in the ocean areas west of South Africa and South
America and land areas of the NH high latitudes. The maximum fire aerosol CRE
occurs in the NH high latitudes in the boreal summer, which results from the large
cloud LWP over the land areas and the large fire OC to BC ratio. Associated with the
strong indirect effects of fire aerosols in the Arctic summer, significant surface
cooling, precipitation reduction, and low-level cloud cover increase are found in these
regions.
Modeled BCS concentrations from the FIRE experiment are evaluated against
observations in Northern China and in the Arctic, and generally agree with the
observations for the mean and median values in the Arctic regions. The high bias of
modeled BCS concentrations in Northern China may not result from the fire BC
because differences in BCS concentrations between FIRE and NOFIRE experiments
are very small in North China. The global annual mean SAE is $0.03 \pm 0.10$ W m$^{-2}$
with the maximum effect in spring (0.12 W m$^{-2}$). The SAE is mainly due to the effect
of fire BC deposit on snow (0.02 W m$^{-2}$) diagnosed from SNICAR with the maximum
effect as large as 0.06 W m$^{-2}$ (when snow is present) in spring.
The fire aerosols reduce the global mean surface air temperature ($T_s$) by $0.03 \pm$
0.03 K and precipitation by $0.01 \pm 0.002$ mm day$^{-1}$. The maximum cooling (~1 K) due
to fire aerosols occurs around 60°N in summer, and a suppression of precipitation
(~0.1 mm day$^{-1}$) is also found there. The strong cooling is a result of the strong
indirect effects (-15 W m$^{-2}$) in the land areas of the Arctic regions (60°N to 90°N).
In our study, the global radiative effect of fire aerosols is estimated from



simulations performed with the 4-mode version Modal aerosol module (MAM4) [*Liu*
*et al.*, 2016], daily fire emissions with prescribed vertical emission profiles, and
higher model resolution (0.9° by 1.25°) compared to earlier modeling studies of fire
aerosols [*Tosca et al.*, 2013; *Ward et al.*, 2012]. In their studies, the GFED fire
aerosol emissions were increased by a factor of 1-3 depending on regions to match the
observed AOD. In our study, we do not apply the scaling factor to the fire aerosol
emissions. Our global annual mean DRE of fire aerosols ($0.155 \pm 0.01$ W m$^{-2}$) is,
however, close to 0.18 W m$^{-2}$ in *Tosca et al.* [2013] and 0.13 W m$^{-2}$ in *Ward et al.*
[2012]. The similar fire aerosol DRE from our study but with smaller fire emissions
than these previous studies can result from (1) the use of MAM4 in our study which
more realistically represents the external/internal mixing of BC with other soluble
aerosol species; (2) the more accurate estimation of DRE of fire aerosols in the
presence of low-level clouds with the method of *Ghan* [2013]; and (3) the inclusion of
vertical emissions of fire aerosols, which allows more efficient transport of fire
aerosols from sources. The CRE due to fire aerosols in our study ($-0.70 \pm 0.05$ W m$^{-2}$)
is smaller than -1.64 W m$^{-2}$ in *Ward et al.* [2012] due to the lower fire POM emissions
used in this study compared to *Ward et al.* [2012]. We note that the model still
underestimates observed AODs (mostly within a factor of 2) at the sites
predominantly influenced by biomass burning aerosols during the fire season, which
implies that the fire aerosol radiative forcing can be stronger than estimated in this
study.



**Acknowledgments**

This work is supported by the Office of Science of the US Department of Energy

(DOE) as the NSF-DOE-USDA Joint Earth System Modeling (EaSM) Program and

the National Natural Science Foundation of China (NSFC) under Grant No.41505062.

The Pacific Northwest National Laboratory is operated for the DOE by the Battelle

Memorial Institute under contract DE-AC05-76RL01830. The authors would like to

acknowledge the use of computational resources (ark:/85065/d7wd3xhc) at the

NCAR-Wyoming Supercomputing Center provided by the National Science

Foundation and the State of Wyoming, and supported by NCAR's Computational and

Information Systems Laboratory. The fire emission data were obtained from the

Global Fire Emissions Database (GFED, http://www.globalfiredata.org). The

AERONET data were obtained from http://aeronet.gsfc.nasa.gov. We thank Xiangjun

Shi for the help with processing the AERONET data.

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



Table 1. Numerical experiments and associated fire aerosol emissions in each experiment.

| Experiment | Ensembles | Fire BC | Fire POM | Fire SO$_2$ |
|---|---|---|---|---|
| FIRE | 10 | On | On | On |
| NOFIRE | 10 | Off | Off | Off |
| NOFIREBC | 10 | Off | On | On |
| NOFIREPOM | 10 | On | Off | On |
| FIRE_BBFFBF | 1 | On | On | On |



Table 2. Global, tropics (25°S to 25°N) and Arctic (60°N to 90°N) annual mean fire aerosol (POM and BC) burdens (mg m$^{-2}$), fire aerosol AOD, total fire aerosol radiative effect (RE) at TOA (W m$^{-2}$), direct radiative effect (DRE, W m$^{-2}$), cloud radiative effect (CRE, W m$^{-2}$), and surface albedo effect (SAE, W m$^{-2}$), and changes in shortwave and longwave cloud forcings (W m$^{-2}$), cloud liquid water path (LWP) (g m$^{-2}$), low-level cloud cover (%), net solar fluxes at surface and in the atmosphere (W m$^{-2}$), surface air temperature (K), and precipitation (total, convective, and large-scale) (mm day$^{-1}$) due to all fire aerosols. Standard deviations about the 10-ensemble means are included.

|  | Global | Tropics (25°S to 25°N) | Arctic (60°N to 90°N) |
|---|---|---|---|
| Fire POM burden | 1.25±0.01 | 1.87±0.01 | 1.70±0.08 |
| Fire BC burden | 0.106±0.001 | 0.17±0.001 | 0.09±0.004 |
| Fire aerosol optical depth | 0.008±0.001 | 0.012±0.001 | 0.007±0.0004 |
| Total radiative effect (RE) | -0.55±0.07 | -0.66±0.09 | -1.35±1.03 |
| Direct radiative effect (DRE) | 0.155±0.01 | 0.172±0.017 | 0.428±0.028 |
| Cloud radiative effect (CRE) | -0.70±0.05 | -0.82±0.09 | -1.38±0.23 |
| Surface albedo effect (over land) | 0.03±0.10 | -0.04±0.06 | 0.09±0.80 |
| Shortwave cloud forcing | -0.43±0.05 | -0.45±0.08 | -1.18±0.22 |
| Longwave cloud forcing | -0.26±0.04 | -0.35±0.07 | -0.04±0.17 |
| Cloud liquid water path | 1.62±0.01 | 1.95±0.13 | 2.59±0.25 |
| Low-level cloud cover | 0.012±0.06 | -0.055±0.05 | 0.46±0.45 |
| Net solar flux at surface | -1.38±0.05 | -1.91±0.12 | -2.27±1.04 |
| Net solar flux in the atmosphere | 0.83±0.03 | 1.25±0.04 | 0.92±0.05 |
| Surface air temperature | -0.03±0.03 | -0.024±0.011 | -0.15±0.2 |
| Total precipitation rate | -0.010±0.002 | -0.016±0.01 | -0.001±0.02 |
| Convective precipitation rate | -0.003±0.002 | -0.001±0.009 | -0.005±0.003 |
| Large-scale precipitation rate | -0.007±0.002 | -0.015±0.003 | 0.004±0.019 |



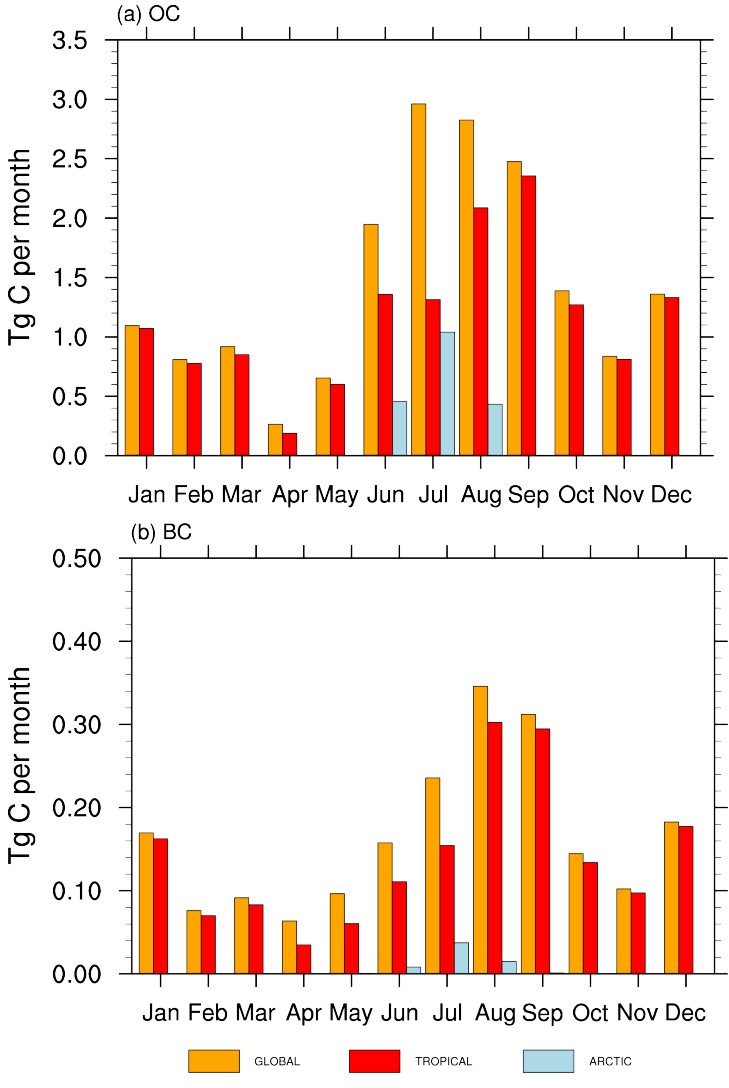

Figure 1. Seasonal variation of GFED monthly fire (a) organic carbon (OC) and (b) black carbon (BC) emissions (Tg C month$^{-1}$) averaged for the period of year 2003 to 2011 in the global, tropical (25°S to 25°N) and Arctic (60°N to 90°N) regions.



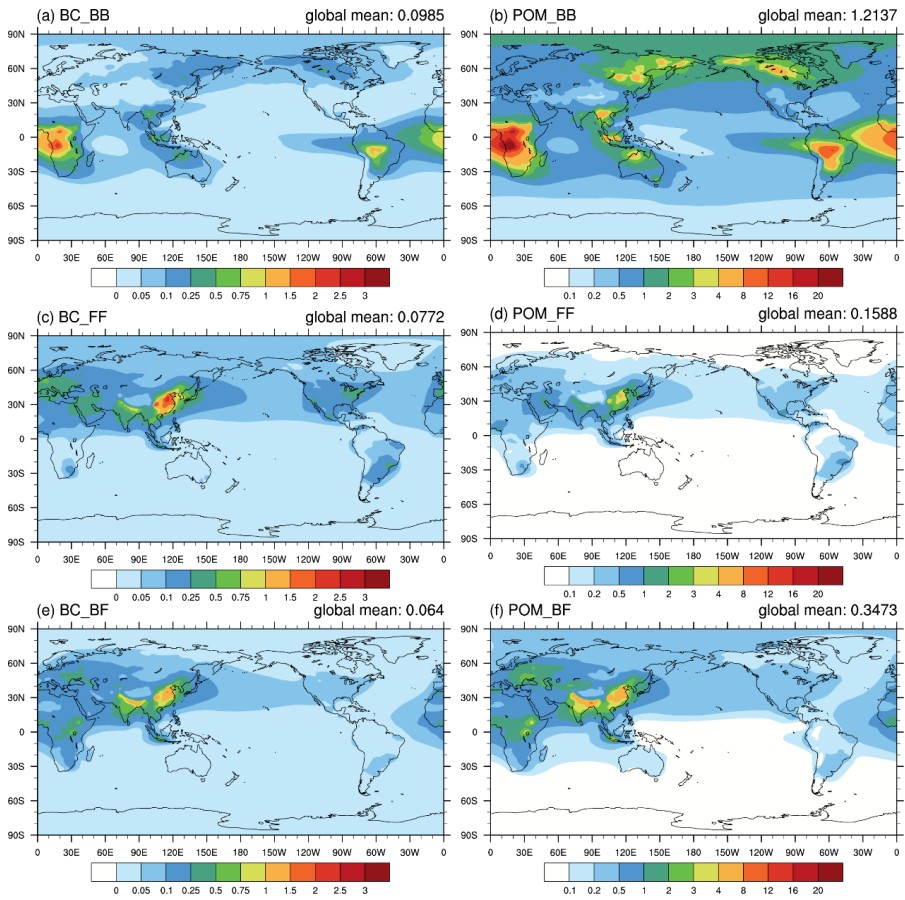

Figure 2. Annual mean (for year 2003-2011) vertically integrated concentrations (units: mg m$^{-2}$) of BC (left) and POM (right) from biomass burning (BB) (upper panel), FF (fossil fuel) (middle panel), and biofuel (BF) (lower panel).





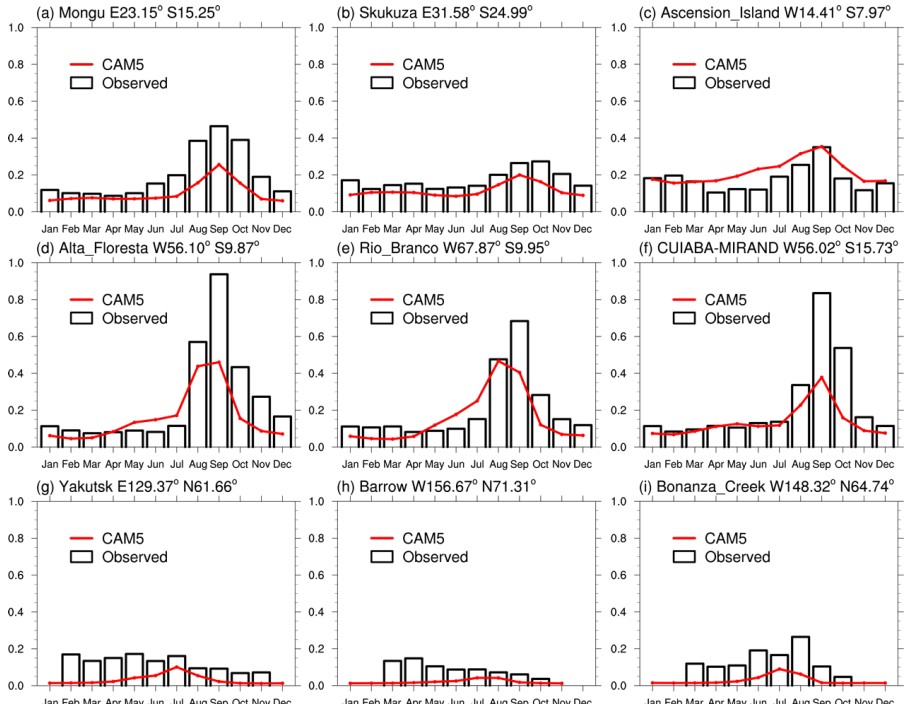

Figure 3. Comparison of modeled seasonal variations of aerosol optical depth (AOD) for the period of 2003-2011 with observations for the same period from the AERONET sites.



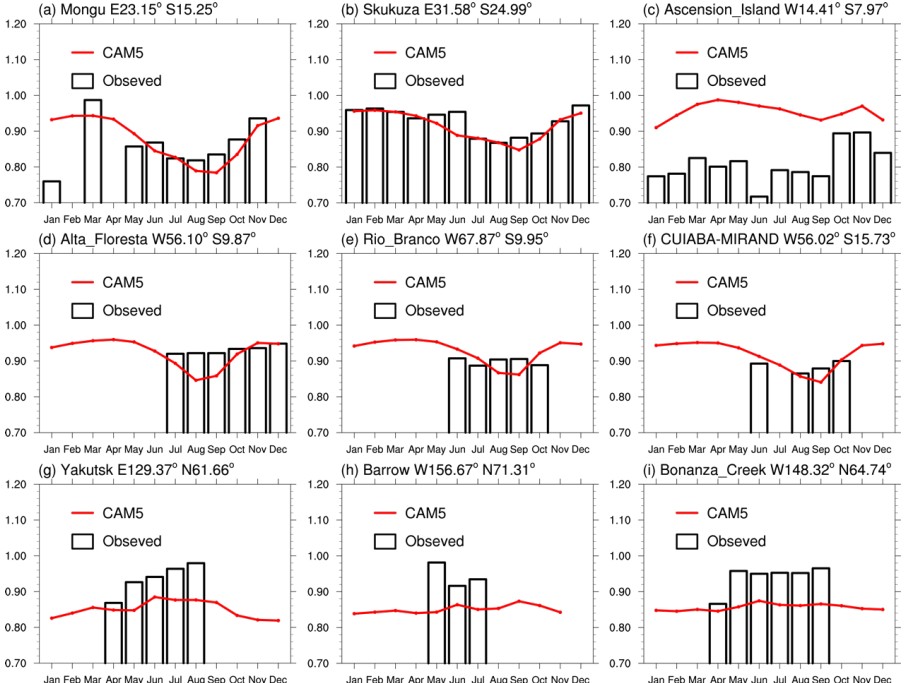

Figure 4. Same as Figure 3, but for the comparison of single scattering albedo (SSA) at 550 nm.



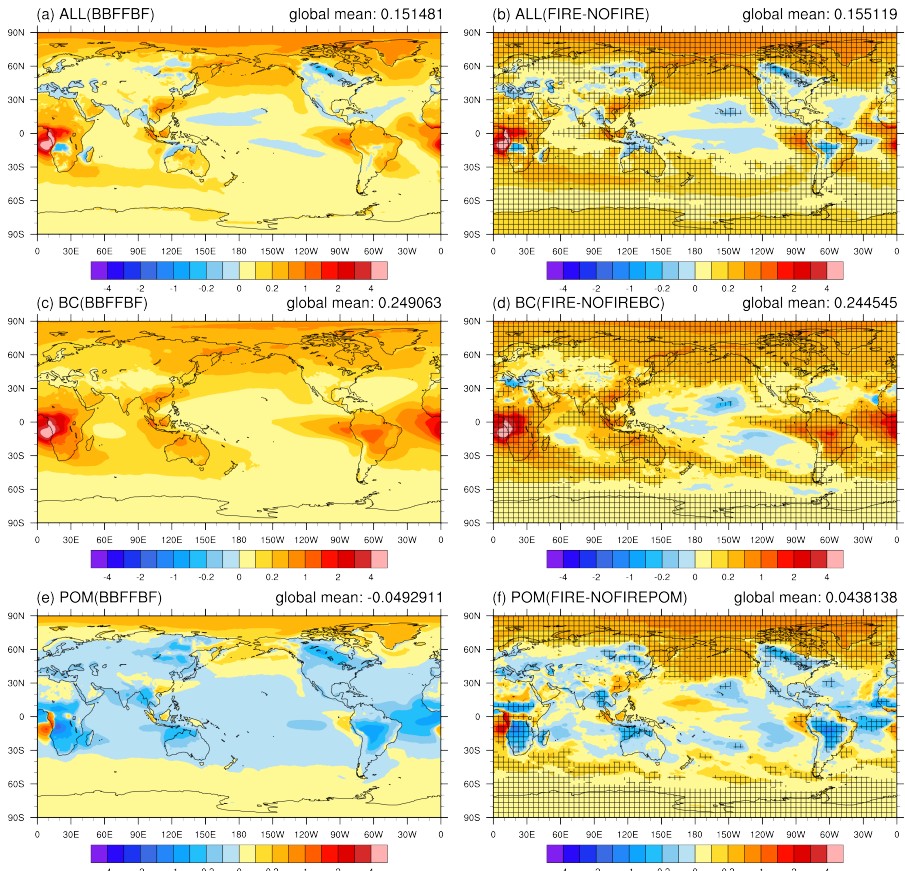

Figure 5. Annual mean direct radiative effect (DRE) (W m$^{-2}$) averaged over the period of 2003-2011 due to (a) all fire aerosols, (c) fire BC, and (e) fire POM estimated with the method of BBFFBF (left panels), and with the method of Ghan [2013] ((b), (d), and (f) in the right panels). The plus signs in Figure 5(b), (d) and (f) denote the regions where the radiative effect is statistically significant at the 0.05 level.



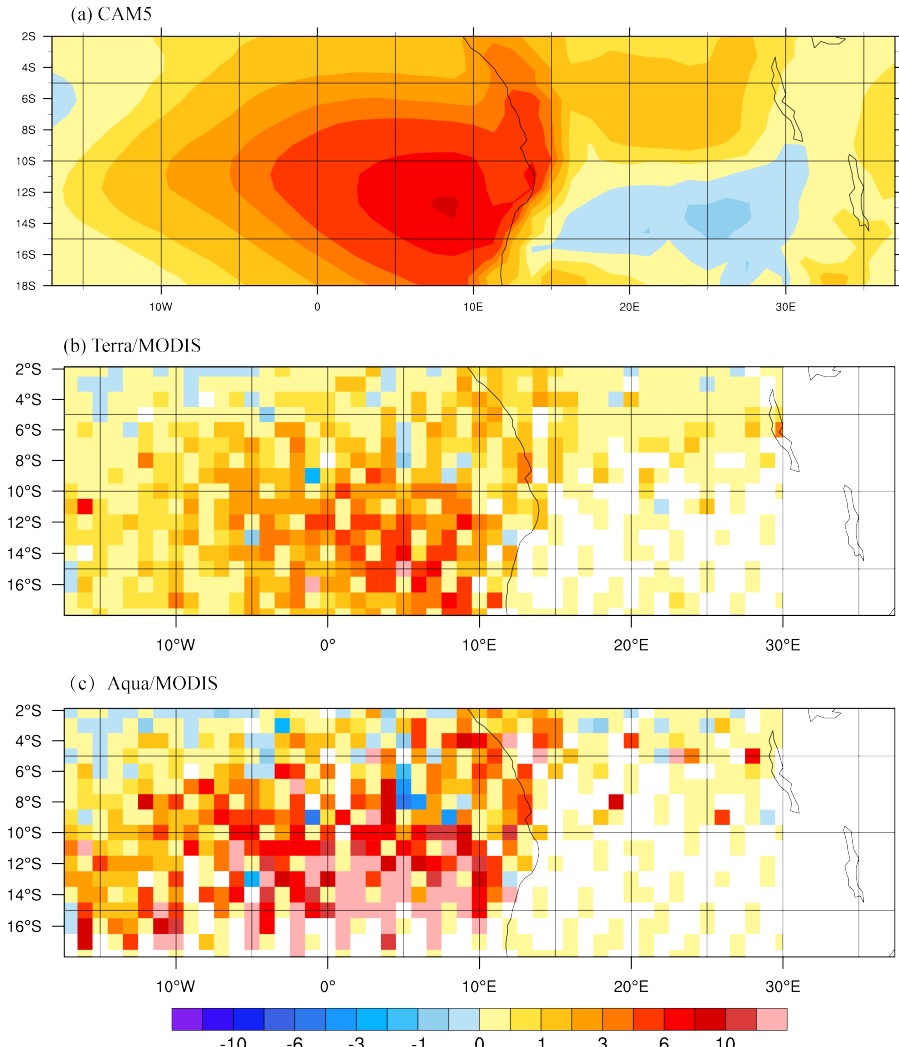

Figure 6. (a) September-October-November (SON) mean fire aerosol direct radiative effect (DRE) (W m$^{-2}$) for the period of 2003-2011 over the Southeast Atlantic Ocean due to all fire aerosols. (b) and (c) are the same as (a), but for the above-cloud aerosol DRE for the period of 2007-2011 estimated using Aqua/MODIS and Terra/MODIS products [*Zhang et al.*, 2014], respectively.



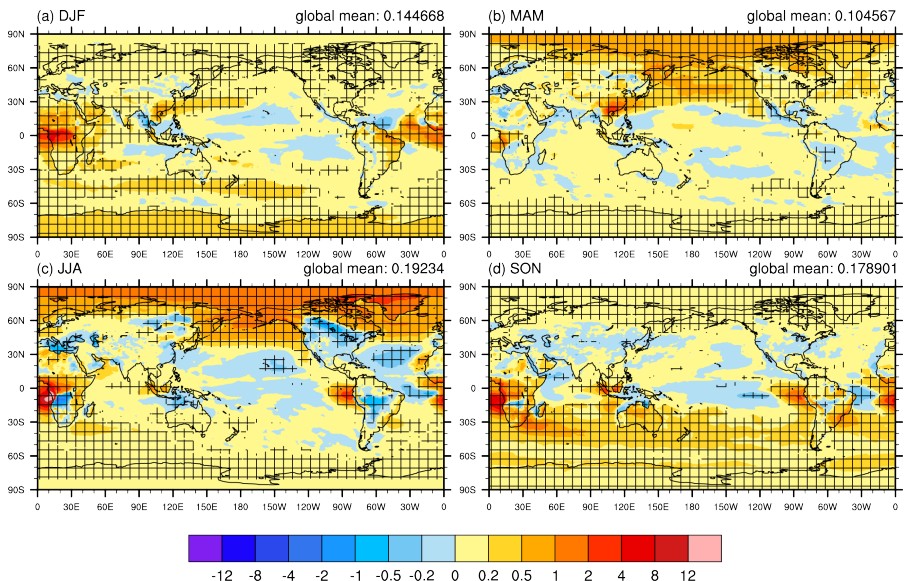

Figure 7. Direct radiative effect (DRE) (W m$^{-2}$) for the period of 2003-2011 due to all
fire aerosols for (a) December-January-February (DJF), (b) March-April-May (MAM),
(c) June-July-August (JJA), and (d) September-October-November (SON). The plus
signs denote the regions where the DRE is statistically significant at the 0.05 level.



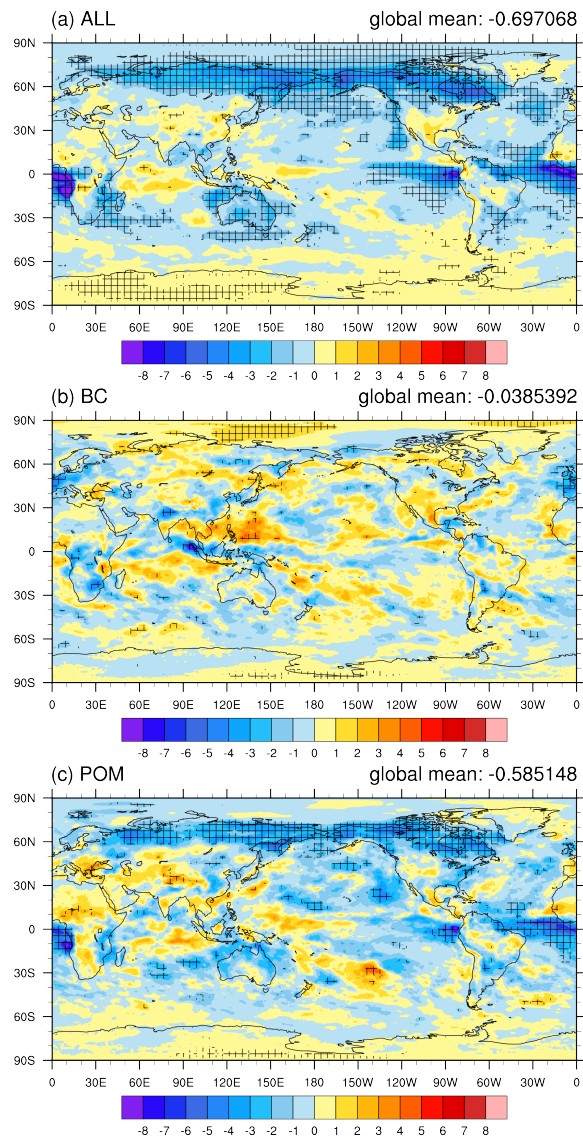

Figure 8. Annual mean cloud radiative effect (CRE) (W m$^{-2}$) averaged over the period of 2003-2011 due to (a) all fire aerosols, (b) fire BC, and (c) fire POM. The plus signs denote the regions where the radiative effect is statistically significant at the 0.1 level.





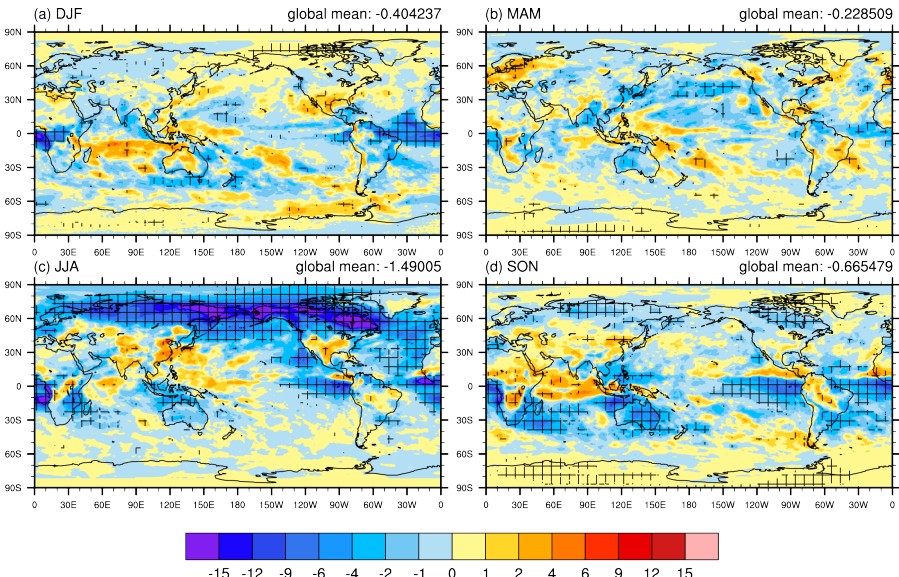

Figure 9. Seasonal variation of cloud radiative effect (CRE) (W m$^{-2}$) due to all fire aerosols for the period of 2003-2011 for (a) December-January-February (DJF), (b) March-April-May (MAM), (c) June-July-August (JJA), and (d) September-October-November (SON). The plus signs denote the regions where the CRE is statistically significant at the 0.05 level.



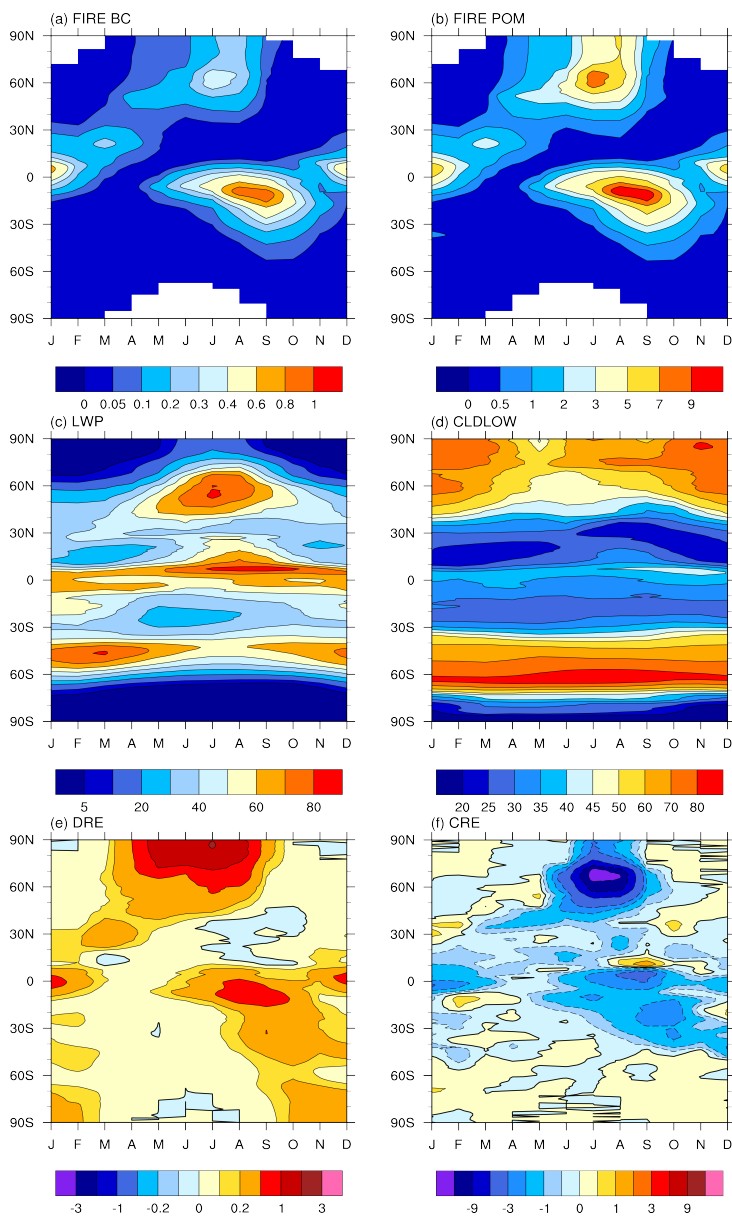

Figure 10. Month-latitude cross sections of zonal mean and monthly (a) vertically-integrated concentrations (mg m$^{-2}$) of fire BC and (b) fire POM, (c) cloud liquid water path (LWP, in g m$^{-2}$), (d) low-level cloud cover (CLDLOW, in %), (e) DRE (W m$^{-2}$), and (f) CRE (W m$^{-2}$) of fire aerosols.





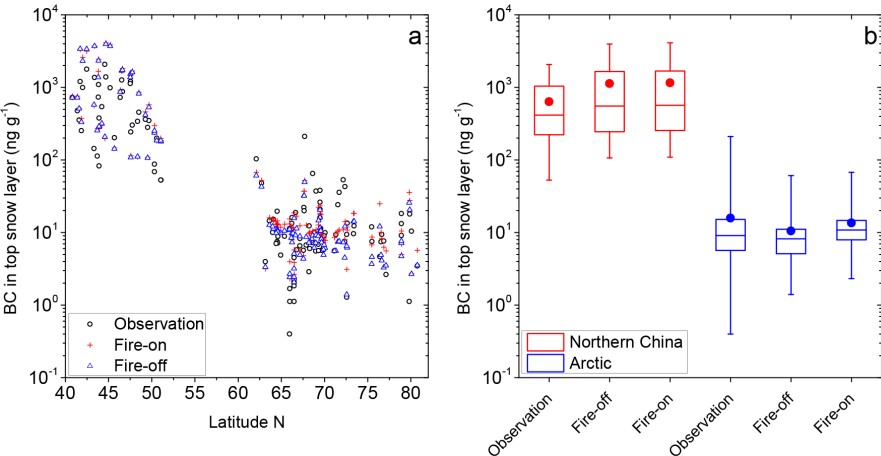

Figure 11. Evaluation of CAM5 simulated black carbon (BC) concentration for the period of 2003-2011 (in ng $g^{-1}$) in the top snow layer against observations in the Arctic (*Doherty et al.*, 2010) and Northern China (*Wang et al.*, 2013b). The top snow layer ranges in thickness from 1 to 3 cm. Configuration of the two CAM5 simulations (FIRE and NOFIRE) is summarized in Table 1. Panel (a) shows the comparisons at different latitudes. The box and whisker plot in panel (b) shows the minimum and maximum value with the bar, the 25th and 75th percentiles with the box, the 50th percentile (i.e., median) by the bar within the box, and the mean value with the dot.





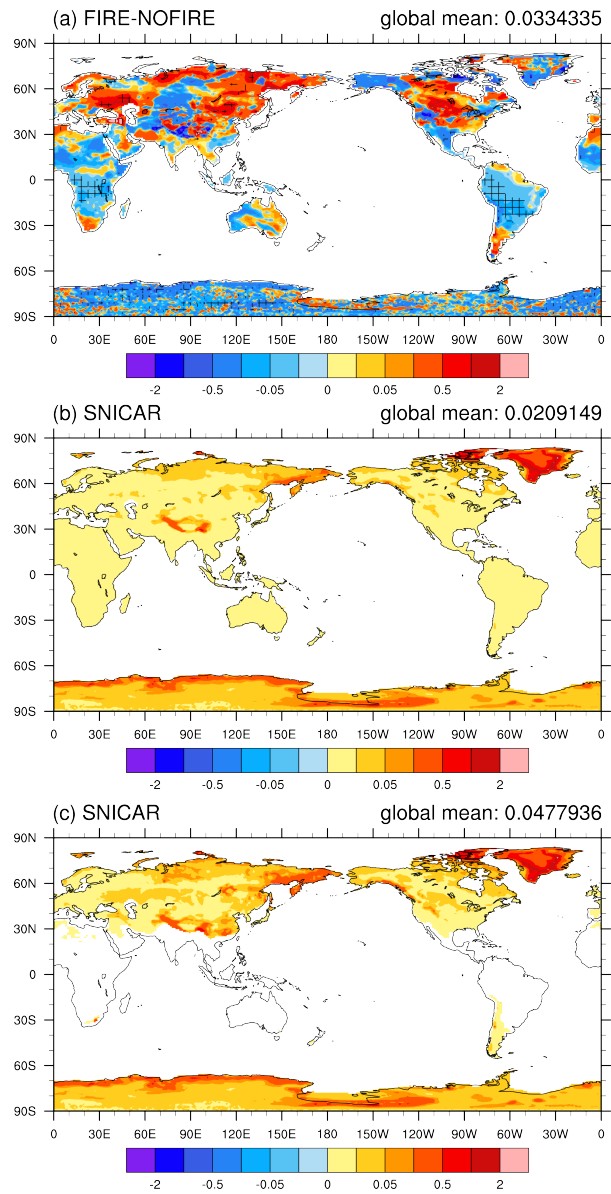

Figure 12. (a) Annual mean surface albedo effect (SAE, W m$^{-2}$) averaged over the period of 2003-2011 of all fire aerosols over land regions, and annual mean surface effect of fire BC-in-snow calculated from SNICAR averaged (b) over all times and (c) only when snow is present. The plus signs in (a) denote the regions where the radiative effect is statistically significant at the 0.1 level.




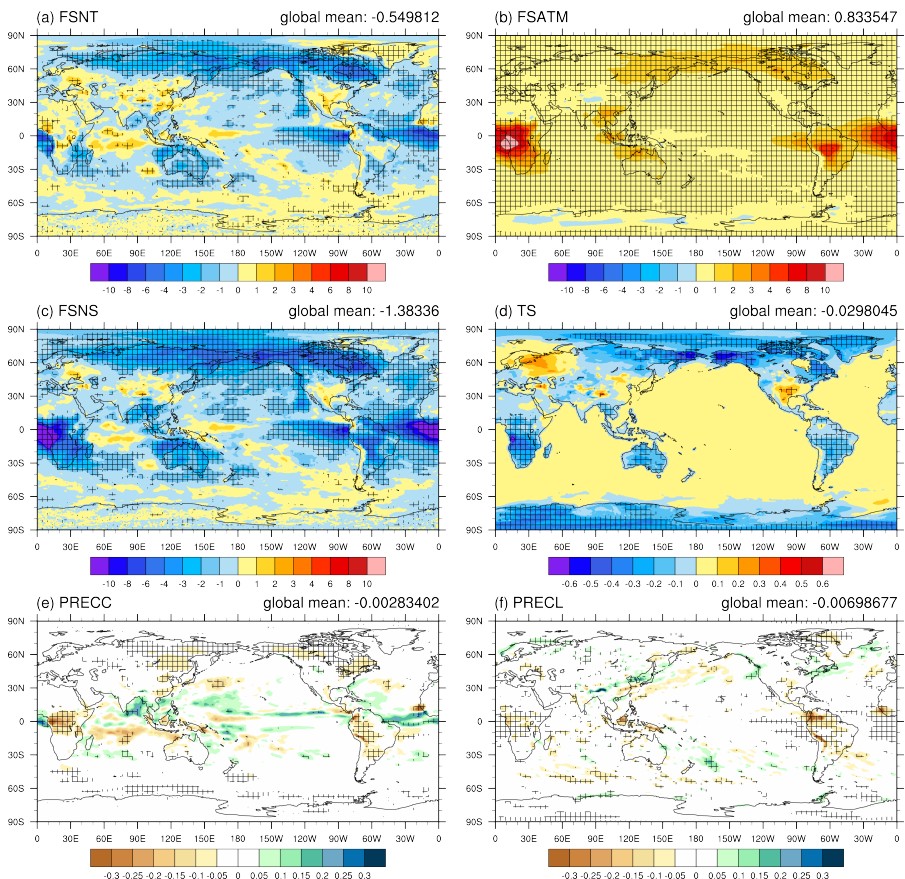

Figure 13. Annual mean net shortwave flux changes (W m$^{-2}$) over the period of 2003-2011 (a) at top of the atmosphere, (b) in the atmosphere, (c) at surface, and changes of (d) surface air temperature (K), (e) convective precipitation (mm d$^{-1}$), and (f) large-scale precipitation (mm d$^{-1}$) due to all fire aerosols. The plus signs denote the regions where the change is statistically significant at the 0.1 level.





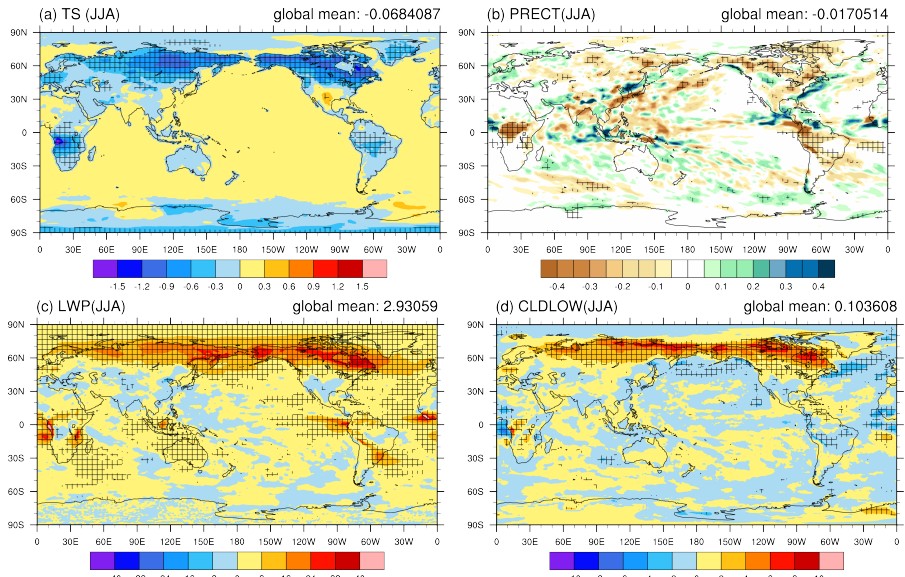

Figure 14. Changes in (a) surface air temperature (K), (b) total precipitation (mm d$^{-1}$), (c) cloud liquid water path (g m$^{-2}$), and (d) low-level cloud cover (%) due to all fire aerosols in the boreal summer (JJA) averaged for the period of 2003-2011. The plus signs denote the regions where the change is statistically significant at the 0.1 level.