# Peer review of "Impacts of Global Open Fire Aerosols on Direct Radiative, Cloud"

_Atmospheric Chemistry and Physics, 2016_

## Referee Comment (RC1) · Anonymous Referee #2 · 8 May 2016

The manuscript of Jiang et al. investigates the impacts of fire-emitted aerosols on the Earth's radiative balance, through direct, indirect, semi-direct, and surface albedo effects. This is pursued via the analysis of some carefully designed global model experiments. The global aerosol direct radiative effect simulated is in agreement with the few previous studies that have focused on this topic, while the indirect effect is found to be somewhat weaker, though still strong. Additionally, the authors discuss the geographical distribution of the fire aerosol effects. The study also briefly describes some implications for climate response, i.e. temperature and precipitation, though that is only done via atmosphere-only simulations.

The manuscript is certainly within the scope of ACP. The topic is important and relatively under-explored, with large uncertainty remaining in how fire emissions affect the Earth's radiative balance and climate. It will be a useful addition to the emerging discussion of the role of fire in the Earth system. The language and structure is ok. I do not have any major concerns, but I have a number of (mostly minor) suggestions that I list below which I believe will improve the manuscript. Following those, I expect that it will be ready for publication.

GENERAL COMMENT:

My only somewhat major comment is that the climate responses explored (i.e. temperature and precipitation) are based on atmosphere-only experiments, and therefore are somewhat incomplete. That does not mean that it is not worth showing the results, but it should be clearly stated that these results come from fixed-SST simulations, and therefore more work will be needed in the future in a coupled framework to understand the role of fire aerosols on climate in a more complete fashion. Adding a few sentences in the abstract, the corresponding section and in the conclusions would be sufficient for clarifying this better.

SPECIFIC COMMENTS:

Page 2, Line 32: Please remove "the".

Page 2, Line 38: Not sure whey a range is indicated both by two numbers and by the +/-

Page 2, Line 39: South Africa -> southern Africa (here and elsewhere in the text).

Page 2, Line 48: Suggest stressing that this effect is small and insignificant.

Page 2, Lines 45-47: Need to clearly mention here that this is inferred from atmosphere-only simulations (and not from full coupled climate simulations).

Page 3, Lines 55-56: Worth citing the review paper by Voulgarakis and Field (2015) here, as it is very relevant.

Page 3, Lines 59-60: Worth citing the paper of Bistinas et al. (2014) here.

Page 3, Lines 61-63: This reads as if this manuscript will fill the gap of knowledge of how fires will change in the future, which is not the case. Please rephrase to something that aligns better with the focus of the manuscript (or you could remove the second part of the sentence entirely).

Page 3, Line 72: Suggest changing "indirect effect" to "indirect effects".

Page 4, Line 76: Suggest removing "the" before "climate change".

Page 4, Lines 76-78: Well, it depends. RE is not always for both anthropogenic and natural. Sometimes we just study anthropogenic or natural RE individually. I suggest rephrasing to "RE represents the instantaneous radiative impact of atmospheric particles on the Earth's energy balance".

Page 4, Lines 78-81: Similarly, RF does not have to always be pre-industrial to present-day. I suggest rephrasing to "...as the change of RE between two different periods, e.g. the pre-industrial and the present-day...", and then change the second half of the sentence accordingly.

Page 5, Line 98: many -> some

Page 7, Lines 147-148: It is mentioned that two methods are presented – worth briefly mentioning them here.

Section 2.1: Is the aerosol interactive with the model's chemistry?

Page 8, Lines 179-180: Any other performance features apart from the Arctic? What about over key biomass burning regions, and what about OC?

Page 9, Lines 194: Suggest changing "climate" to "atmospheric" or "short-term climate" as the SSTs/sea ice are prescribed.

Page 10, Lines 227-228: I may be missing something here, but how can the difference

between F in two simulations that do not involve any aerosols ("clean" and "clean,clear") tell you something about the aerosol-induced cloud radiative effect (CRE)?

Page 11, Line 238: Could add "each time" before "neglecting the. . .".

Page 11, Lines 239: Suggest adding "more direct" between "This" and "method".

Page 12, Line 257: topics -> tropics

Page 13, Line 284: activities -> activity

Page 13, Lines 286-288: If scaling is not applied here, mention it clearly (e.g. ". . .whereas here we do not apply any such scaling").

Page 13, Line 294: trend -> seasonal cycle

Figure 3: Do the selected AERONET sites have data for exactly the same years as the simulation? Not entirely necessary, but needs to be mentioned. Also: Worth mentioning in the caption (also in Fig. 4) that the first row shows sites in southern Africa, the second row sites in South America, and the third row sites in the Arctic.

Page 14, Lines 303-304: There is also a notable early peak. Worth mentioning and perhaps commenting on.

Page 14, Line 306: However, there is too strong a seasonality, it seems? Any explanation?

Page 15, Line 321: Sulfate and OC, right?

Page 15, Line 327: No need for "respectively" here.

Figure 5: There are too many significant figures in the global mean values shown on each panel (also in later figures). Also: With respect to what is statistical significance estimated for the right panels? Interannual variability or ensemble member diversity? Needs to be mentioned here and also in later figures. And why is significance not shown for the left hand panels?

Page 15, Line 332: Why have you chosen to report only the global mean from the BBFFBF method in the text, and not from the one based on Ghan (2013)?

Page 15, Line 336: "The" is not needed.

Page 16, Line 340: "of the tropical regions" -> "of the SH tropical regions"

Figure 6: Is the model panel (a) produced with all-sky values? In fact, was that the case for Figure 5 too?

Figure 7: Which method was used for those maps to be made?

Page 16, Line 354: Define "high latitudes" here. Is it the same definition as the Arctic?

Page 16, Lines 359-360: "there are much less noises from" -> "there is much less noise with"

Page 17, Lines 371-373: Why would it affect BC? Not clear. Explain better.

Page 17, Line 373: it -> one

Page 17, Lines 375-376: global regions -> globe

Page 17, Lines 378-382: Could the authors provide a reference for this mechanism?

Sect. 3.3: Can's some of the cloud changes that lead to indirect effects be a result of dynamical changes due to fire aerosols?

Page 19, Line 418: Please provide reference to support this statement ("Larger...").

Page 19, Lines 420-421: What does "low-level" mean here?

Page 20, Lines 434-435: The higher OC/BC ratio does not seem like a good explanation, as it is mentioned a bit earlier that POM and BC are comparable in the NH and SH.

Page 21, Line 449: I suggest adding "slightly" between "agree" and "better".

Page 21, Line 452: It reads as if you take values from Ghan (2013). Suggest rephrasing.

Page 21, Lines 469-470: Even in tropical areas? Please discuss.

Page 22, Line 484: Instead of "The shortwave flux change in the atmosphere ", I suggest writing "The shortwave atmospheric absorption change", as it is more conventional.

Figure 13a: Clarify to the reader why the values in Fig. 8a are somewhat different to those in Fig. 13a.

Page 23, Lines 505-506: There are also substantial differences with Tosca et al. (2013), especially over tropical oceans, therefore I would add "partly" before "consistent". Also the results over southern Africa are consistent with the recent findings of Hodnebrog et al. (2016), which the authors can mention.

Page 23, Line 511: After this line, I suggest that you add a statement clearly stating that these results do not represent the complete impact of fire emitted aerosols on temperature and (especially) precipitation, since the climate system has not been allowed to fully respond (SSTs are fixed).

Page 24, Line 519: effect -> effective

Page 26, Lines 575-579: Again, I suggest reminding the reader that these do not represent the full climate responses, given the atmosphere-only nature of the experiments.

Page 27, Lines 596-597: Is the difference in emitted POM between the two studies equivalent (in size) to the difference in the CRE?

REFERENCES:

Bistinas, I., Harrison, S. P., Prentice, I. C., and Pereira, J. M. C. (2014), Causal relationships versus emergent patterns in the global controls of fire frequency, Biogeosciences, 11, 5087-5101, doi:10.5194/bg-11-5087-2014.

[Figure]

Hodnebrog et al. (2016), Local biomass burning is a dominant cause of the observed precipitation reduction in southern Africa, Nature Communications, doi:10.1038/ncomms11236.

Voulgarakis, A., and R.D. Field, 2015: Fire influences on atmospheric composition, air quality, and climate. Curr. Pollut. Rep., 1, no. 2, 70-81, doi:10.1007/s40726-015-0007-z.

---

## Referee Comment (RC2) · Anonymous Referee #1 · 7 Jun 2016

Review of "Impacts of global wildfire aerosols on direct radiative, cloud and surface-albedo forcings simulated with CAM5," by Jiang et al.

This paper examines the global and regional radiative forcings by black carbon and organic carbon aerosols from open fires. The authors use the NCAR Community Atmosphere Model version 5.3 (CAM5) with the four-mode version of the modal aerosol module (MAM4) and employ two methods to calculate forcing. In one method, they follow Ghan et al. (2013), which may produce a more robust estimate of forcing. In the second method, they follow a more traditional approach. The authors find that top-of-atmosphere (TOA) forcing from aerosol-cloud interactions dominates the total global forcing (-0.70 Wm$^{-2}$). When aerosol-radiation interactions and aerosol effects on snow are also considered, the global annual mean forcing from open fire aerosols is -0.55 Wm$^{-2}$. The authors also estimate the climate impacts of fire aerosols.

The paper leads to no startling new conclusions, but may provide a more accurate estimate of the global and regional climate impacts of aerosols from open fires. The paper should be revised in response to the major criticisms and resubmitted.

**Major criticisms.**

1. The paper needs to make more clear what is new in the results, or why this approach represents a substantial improvement over previous results. Central to this paper should be the answer to this question: Why does this research give us greater confidence in our knowledge of the effects of fire aerosols on climate?

In Lines 147-150, the text lists a few improvements, but supplies little elaboration. The improvements are: (a) higher spatial resolution, (b) use of the latest CAM5 model with updated MAM4, (c) calculation of daily instead of monthly fire emissions, and (d) use of an alternative methodology to calculate radiative forcings of aerosols (Ghan 2013). It's not clear why the relatively small increase in spatial resolution would lead to better results, or why calculation of daily instead of monthly fires matters. Almost no information on the updates in MAM4 is given or what difference they make for forcing calculations. A detailed explanation of the benefits of the Ghan (2013) method over other methods is absent.

2. The paper uses outdated terms to describe radiative forcing by aerosol, and does not adequately describe what adjustments to the model meteorology have been allowed in the forcing calculations. Following IPCC AR5, the authors should use the terms aerosol-radiation interactions (AR1), aerosol-cloud interactions (ACI), and forcings due to surface albedo changes (Boucher et al., 2014; Myhre et al., 2014). ACI in the IPCC framework includes the effects of aerosols on cloud droplet number, cloud lifetime and takes into account the "semi-direct effect" of absorbing aerosols. The ACI category of forcings is useful as it makes it unnecessary to distinguish between the sometimes competing effects of aerosols on clouds.

The authors should further state whether they calculated radiative forcings (RF) or effective radiative forcings (ERF), which take into account the rapid adjustments to a range of meteorological variables. If these are ERFs (and they seem to be), the authors need to make

clear what meteorological variables they allowed to adjust. The authors should emphasize in the abstract and conclusions that the forcings they report are relative to the case of no fires, and not to conditions in 1750s.

3. It's not clear why the paper does not consider the effects of fire aerosols on sea ice albedo. Is this not an important forcing term? Also the authors neglect the issue of brown carbon, which has recently been suggested as a main component of primary organic matter (POM) in fire plumes (Feng et al., 2013). MAM4 may not be capable of simulating brown carbon, and this should be acknowledged.

4. The authors report a large number of changes in global mean variables without giving uncertainty ranges or stating which changes are statistically significant. Given that many of the variables have been calculated using an ensemble of simulations, uncertainties should be easy to calculate.

**Other criticisms.**

Title: Given the distribution of fires in Figure 2, it looks like the authors include agricultural fires in their analysis, and so the term "wildfire" should be changed to "open fires."

Abstract. The abstract should state the time period under investigation. Also large regional forcings should be quantified, as they could have importance for regional climate.

Introduction. The introduction is too long. The first paragraph should make clear exactly what problem is being considered, and it should succinctly explain why this investigation represents a major improvement over past research. Throughout the introduction, many old references brought up – e.g., Chuang et al. (2002) or IPCC AR4. The authors should condense the introduction and focus on Chapters 7 and 8 in AR5 and subsequent papers – e.g., Myhre and Samset (2015), Chakrabarty et al. (2014), and many others. Missing from the introduction is a discussion of the radiative effects of organic vs black carbon.

Line 174. The authors state that MAM4 "significantly increases (and improves) the BC concentrations in the Arctic…." Why does inclusion of the primary carbon mode in MAM4 improve the treatment of microphysical aging of BC? How did the authors decide that inclusion of this mode "significantly" improves the BC simulation? By what measure? Elsewhere the authors state that MAM4 "realistically represents the external/internal mixing of BC" (Line 578). But no detail is given about these improvements.

Section 2.3. See major criticism #2 above. Please rewrite using IPCC AR5 convention for describing forcings.

Results. The results section rambles. The authors should decide which are the key results and provide more detailed explanations of the mechanisms driving these results. Also, the statistical significance of results should be given, where possible. Since the authors performed an ensemble of simulations, many results can be reported with one standard deviation uncertainty. For

example, what is the uncertainty of the forcings calculated following Ghan 2013? Is the -0.03°C temperature effect of fire aerosols statistically significant?

Finally, the forcings calculated for specific regions should be compared to recently published estimates – e.g. Brieder et al. (2014) for the Arctic and Sena and Artaxo (2015) for South America.

Line 241. Here and elsewhere. It is not clear whether the fires examined in this study include agricultural fires such as those in Equatorial Asia and South America.

Lines 276-on. The text should state whether the modeled AOD includes aerosol from all sources, not just fires.

Line 311. The text states, "Although MAM4 increases the column burdens of POM and BC by up to 40% in many remote regions compared to MAM3…." Why does this large increase occur?

Line 338. Text should be more clear about how clouds amplify the forcing of BC.

Line 343. Why is the forcing estimated from Terra different from that of Aqua?

Line 346. There is no mention here or elsewhere about the effect of solar zenith angle on radiative forcing at high latitudes, particularly the Arctic.

Line 349. Here and elsewhere, the authors should take care with the terms "summer" and "autumn" when referring to the Southern Hemisphere.

Line 354. "noises" Please fix English.

Line 364. The text states: It is not clear why removal of POM in the simulation affects BC concentrations. If indeed this is what happens, then the Ghan method for calculating forcing should not be used for individual fire components.

Line 379. See above comment.

Cloud radiative forcing section. Please see major criticism #2. Also, this section should provide discussion of why the forcing due to ACI is stronger in some regions compared to others.

Line 411. The text should state why larger cloud liquid water path leads to stronger forcing due to ACI.

Section on surface snow albedo forcing. Why are forcings due to BC deposition on sea ice not considered? The section seems misnamed, since forcings on all light colored surfaces are seen in Figure 12.

The forcings on surface albedo calculated with the Ghan 2013 method look suspiciously high over low latitudes (Figure 12). The authors should comment on these high values – e.g., +0.5 Wm$^{-2}$ over parts of the U.S. south. Are these results comparable to those from SNICAR?

Figure 12b reveals no significant differences in forcings for the fire vs no-fire cases over the Arctic or north China. The authors should acknowledge this. Given the results from SNICAR, it seems that the only region that might show a significant impact of fire aerosols on surface albedo is Greenland and the very northern reaches of Canada.

Line 458. It sounds like snow melting is one of the rapid meteorological adjustments allowed to occur in the forcing calculation. Is this correct?

Section on the fire aerosol effects on shortwave radiation, global temperature and precipitation. Here the statistically significance and the uncertainties of global results should be stated. If the global mean changes of some variables are not statistically significant, then that should be made clear.

Discussion section. Again the authors should stress the key points and put them in context of other new studies besides just Ward 2012 and Tosca 2013. What exactly is new in this study? Limitations and uncertainties of the study should be discussed – i.e., what are the shortcomings of the approach used here?

**Tables and Figures.**
There are too many Figures. Decide what is important and put rest in a supplement.

Captions should be stand-alone so that the browsing reader can understand what is being shown. Unusual acronyms should be explained.

Units in Table 2 should be within the table, not in the caption.

Uncertainty ranges should be included in Table 2, and significant changes shown in boldface.

Text on all legends should be large enough to read. The latitude and longitude labels on the global maps can be eliminated for a cleaner, less cluttered appearance.

Global mean values should be reported to 2-3 significant digits.

Figures 4 and 5 should include error bars.

Figure 7. What does white space represent?

Figure 14. Replace acronyms above the panels with standard English terms.

**References.**

Boucher, O., D. Randall, P. Artaxo, C. Bretherton, G. Feingold, P. Forster, V.-M. Kerminen, Y. Kondo, H. Liao, U. Lohmann, P. Rasch, S.K. Satheesh, S. Sherwood, B. Stevens and X.Y. Zhang, 2013: Clouds and Aerosols. In: Climate Change 2013: The Physical Science Basis. Contribution of Working Group I to the Fifth Assessment Report of the Intergovernmental Panel on Climate Change [Stocker, T.F., D. Qin, G.-K. Plattner, M. Tignor, S.K. Allen, J. Boschung, A. Nauels, Y. Xia, V. Bex and P.M. Midgley (eds.)]. Cambridge University Press, Cambridge, United Kingdom and New York, NY, USA.

Breider, T.J., L.J. Mickley, D.J. Jacob, Q. Wang, J.A. Fisher, R.Y.-W. Chang, and B. Alexander (2014), Annual distributions and sources of Arctic aerosol components, aerosol optical depth, and aerosol absorption, J. Geophys. Res. Atmos., 119, 4107-4124.

Chakrabarty, R. K., N. D. Beres, H. Moosmüller, S. China, C. Mazzoleni, M. K. Dubey, L. Liu, and M. I. Mishchenko (2014), Soot superaggregates from flaming wildfires and their direct radiative forcing, Sci. Rep., 4, doi:10.1038/srep05508.

Feng, Y., V. Ramanathan, and V. R. Kotamarthi (2013), Brown carbon: a significant atmospheric absorber of solar radiation?, Atmos. Chem. Phys., 13(17), 8607–8621, doi:10.5194/acp-13-8607-2013.

Myhre, G., D. Shindell, F.-M. Bréon, W. Collins, J. Fuglestvedt, J. Huang, D. Koch, J.-F. Lamarque, D. Lee, B. Mendoza, T. Nakajima, A. Robock, G. Stephens, T. Takemura and H. Zhang, 2013: Anthropogenic and Natural Radiative Forcing. In: Climate Change 2013: The Physical Science Basis. Contribution of Working Group I to the Fifth Assessment Report of the Intergovernmental Panel on Climate Change [Stocker, T.F., D. Qin, G.-K. Plattner, M. Tignor, S.K. Allen, J. Boschung, A. Nauels, Y. Xia, V. Bex and P.M. Midgley (eds.)]. Cambridge University Press, Cambridge, United Kingdom and New York, NY, USA.

Myhre, G., and B.H. Samset (2015), Standard climate models radiation codes underestimate black carbon radiative forcing, Atmos. Chem. Phys., 15, 2883–2888.

Sena, E.T., and P. Artaxo (2015), A novel methodology for large-scale daily assessment of the direct radiative forcing of smoke aerosols, Atmos. Chem. Phys., 15, 5471–5483.

---

## Author Comment (AC1) · 5 Aug 2016

We thank the reviewer for his/her careful reviews and helpful comments. The manuscript has been revised accordingly and our point-by-point responses are provided below. (Reviewer's comments are in italic and the responses in standard font).

**Reviewer #2**

*General comments*

*My only somewhat major comment is that the climate responses explored (i.e. temperature and precipitation) are based on atmosphere-only experiments, and therefore are somewhat incomplete. That does not mean that it is not worth showing the results, but it should be clearly stated that these results come from fixed-SST simulations, and therefore more work will be needed in the future in a coupled framework to understand the role of fire aerosols on climate in a more complete fashion. Adding a few sentences in the abstract, the corresponding section and in the conclusions would be sufficient for clarifying this better*

Reply: Following the reviewer's comment, we have added the statement in the abstract, the corresponding section and in the conclusions that these results (i.e., climate responses) are based on atmosphere-only experiments with fixed-SSTs, and more work using a coupled atmosphere-ocean model will be needed in the future to understand the role of fire aerosols on climate in a more complete fashion.

*Specific Comments:*

*Page 2, Line 32: Please remove "the".*
Reply: Done.

*Page 2, Line 38: Not sure whey a range is indicated both by two numbers and by the +/-*
Reply: We changed to "-0.05 W m$^{-2}$ and 0.04±0.01 W m$^{-2}$, respectively based on two calculation methods". The first number does not have an uncertainty range, since it is derived from a clean calculation.

*Page 2, Line 39: South Africa -> southern Africa (here and elsewhere in the text).*
Reply: Done here and elsewhere in the text.

*Page 2, Line 48: Suggest stressing that this effect is small and insignificant.*
Reply: Done. We added this information to the abstract.

*Page 2, Lines 45-47: Need to clearly mention here that this is inferred from atmosphere-only simulations (and not from full coupled climate simulations).*
Reply: We added this information in the abstract.

*Page 3, Lines 55-56: Worth citing the review paper by Voulgarakis and Field (2015) here, as it is very relevant.*
Reply: Done.

*Page 3, Lines 59-60: Worth citing the paper of Bistinas et al. (2014) here.*
Reply: Done.

*Page 3, Lines 61-63: This reads as if this manuscript will fill the gap of knowledge of how fires will change in the future, which is not the case. Please rephrase to something that aligns better with the focus of the manuscript (or you could remove the second part of the sentence entirely).*
Reply: Following the reviewer' comment, we removed the second part of the sentence.

*Page 3, Line 72: Suggest changing "indirect effect" to "indirect effects"*
Reply: Done.

*Page 4, Line 76: Suggest removing "the" before "climate change".*
Reply: Done.

*Page 4, Lines 76-78: Well, it depends. RE is not always for both anthropogenic and natural. Sometimes we just study anthropogenic or natural RE individually. I suggest rephrasing to "RE represents the instantaneous radiative impact of atmospheric particles on the Earth's energy balance"*

Reply: Thanks. We revised the sentence as the reviewer suggests.

*Page 4, Lines 78-81: Similarly, RF does not have to always be pre-industrial to presentday. I suggest rephrasing to ". . .as the change of RE between two different periods, e.g. the pre-industrial and the present-day. . .", and then change the second half of the sentence accordingly.*

Reply: Thanks. We have revised the sentence accordingly.

*Page 5, Line 98: many -> some*
Reply: Done.

*Page 7, Lines 147-148: It is mentioned that two methods are presented – worth briefly mentioning them here.*

Reply: Thanks for the suggestion. We added the following sentence to briefly mention the two methods: "One method estimates the DRE with different model simulations

[*Ghan*, 2013], and the other one calculates the DRE directly by multiple diagnostic radiation calls in a single simulation.*"*

*Section 2.1: Is the aerosol interactive with the model's chemistry?*

Reply: The secondary aerosol, e.g., sulfate is produced from the model's gas and aqueous sulfur chemistry. The version of the model we are using in this study does not include a full-chemistry mechanism, and the oxidants (e.g., OH, $HO_2$ and $O_3$) are prescribed [*Liu et al.*, 2012].

*Page 8, Lines 179-180: Any other performance features apart from the Arctic? What about over key biomass burning regions, and what about OC?*

Reply: Compared to MAM3, MAM4 increases the concentrations of BC and POM in most global regions. The increase is the strongest over the remote regions (e.g., oceans and Arctic) and relatively small over the land source regions [*Liu et al.*, 2016]. We modified the sentence to include more discussion of performance features.

*Page 9, Lines 194: Suggest changing "climate" to "atmospheric" or "short-term climate" as the SSTs/sea ice are prescribed.*

Reply: Following the reviewer's comment, we changed "climate" to "atmospheric".

*Page 10, Lines 227-228: I may be missing something here, but how can the difference between F in two simulations that do not involve any aerosols ("clean" and "clean, clear") tell you something about the aerosol-induced cloud radiative effect (CRE)?*

Reply: Typically, the aerosol-induced CRE is estimated by the difference of the shortwave cloud forcings ($\Delta$ SWCF, or $\Delta$ (F- $F_{clear}$)) between two simulations. With this method, however, the absorbing aerosols above clouds will produce a positive direct forcing and induce a bias in estimated CRE (Ghan, 2013). Ghan (2013) indicates that CRE be calculated under the clean conditions (i.e., no aerosol *direct* effects). The clean conditions are not meant to have no aerosols in the control simulation, but to have no aerosols in the diagnostic radiation call in the same control simulation. So the SWCF in clean conditions ($\Delta$ SWCF $_{clean}$, or $\Delta$ ($F_{clean} - F_{clean,clear}$)) is used to estimate the CRE.

As we define in the text, "$F_{clean}$ is the radiative flux at TOA calculated from a *diagnostic radiation call* in the same control simulations, but neglecting the scattering and absorption of solar radiation by aerosols."

*Page 11, Line 238: Could add "each time" before "neglecting the. . .".*
Reply: Done.

*Page 11, Lines 239: Suggest adding "more direct" between "This" and "method".*
Reply: Done.

*Page 12, Line 257: topics -> tropics*
Reply: Done. Thanks.

*Page 13, Line 284: activities -> activity*
Reply: Done.

*Page 13, Lines 286-288: If scaling is not applied here, mention it clearly (e.g. ". . .whereas here we do not apply any such scaling").*

Reply: Done. We added the following words: ", whereas here we do not apply any such scaling."

*Page 13, Line 294: trend -> seasonal cycle*
Reply: Done.

*Figure 3: Do the selected AERONET sites have data for exactly the same years as the simulation? Not entirely necessary, but needs to be mentioned. Also: Worth mentioning in the caption (also in Fig. 4) that the first row shows sites in southern Africa, the second row sites in South America, and the third row sites in the Arctic.*

Reply: We downloaded the AERONET data at the selected sites for exactly the same years as the simulation. However, the selected AERONET sites have missing data for some periods of the model simulation, as shown in Figures 3 and 4. We mentioned this in the revision.
We now added the site information to the figure caption.

*Page 14, Lines 303-304: There is also a notable early peak. Worth mentioning and perhaps commenting on.*

Reply: Yes, the modeled AOD shows a notable early peak before the fire season, especially for Alta Floresta and Rio Branco, which could be due to the model overestimation of fire emission in this period. We mentioned this in the revision.

*Page 14, Line 306: However, there is too strong a seasonality, it seems? Any explanation?*

Reply: Yes, the modeled SSA is too low during the fire season and exhibits too strong a seasonality. It implies that the model underestimation of scattering aerosols (e.g., POM) may be more severe than that of BC during the fire season.

*Page 15, Line 321: Sulfate and OC, right?*

Reply: Yes. We changed to "e.g., sulfate and POM".

*Page 15, Line 327: No need for "respectively" here.*
Reply: Thanks. We removed "respectively" here.

*Figure 5: There are too many significant figures in the global mean values shown on each panel (also in later figures). Also: With respect to what is statistical significance estimated for the right panels? Interannual variability or ensemble member diversity? Needs to be mentioned here and also in later figures. And why is significance not shown for the left hand panels?*

Reply: Following the reviewer's comment, we reduced the number of figures in the global mean values shown on each panel (also in later figures) in the revised manuscript. The statistical significance test is applied to the results using the Ghan (2013) method, because DRE from this method is calculated as the radiative flux difference between two model simulations. Therefore, the difference is not only from the DRE of fire aerosols, but also from the model internal variability which includes both the interannual variability (2003-2011) and the ensemble member diversity (10 members). We mentioned this in the revised manuscript.

The statistical significance test is not applied to the BBFFBF method (shown in the left hand panels). The reason is that DRE using this method is calculated as the radiative flux difference between the control run and diagnostic radiation calls in each model time step, which ensures that the climate background (e.g., clouds) is exactly the same between the control run and diagnostic calls.

*Page 15, Line 332: Why have you chosen to report only the global mean from the BBFFBF method in the text, and not from the one based on Ghan (2013)?*

Reply: Actually here the global mean ($0.155\pm0.01$ W m$^{-2}$) is from the Ghan (2013) method. With the Ghan (2013) method, the radiative effects including DRE, CRE and SAE of fire aerosols can be estimated, while the BBFFBF method only estimates DRE. We added a note in the revised manuscript that the two methods give very similar results for DRE of all fire aerosols, and thus we will report the DRE of all fire aerosols with the Ghan [2013] method.

*Page 15, Line 336: "The" is not needed.*
Reply: removed.

*Page 16, Line 340: "of the tropical regions" -> "of the SH tropical regions"*
Reply: Done.

*Figure 6: Is the model panel (a) produced with all-sky values? In fact, was that the case for Figure 5 too?*
Reply: It is DRE in the all-sky condition. This is also the case for Figure 5.

*Figure 7: Which method was used for those maps to be made?*
Reply: It is from the method of Ghan (2013). After a comparison with method BBFFBF, the DRE due to all fire aerosols estimated with Ghan (2013) is used in the rest of the paper. We added a note in the revised manuscript.

*Page 16, Line 354: Define "high latitudes" here. Is it the same definition as the Arctic?*
Reply: We changed the "high latitudes" to "Arctic regions".

*Page 16, Lines 359-360: "there are much less noises from" -> "there is much less noise with"*
Reply: Done.

*Page 17, Lines 371-373: Why would it affect BC? Not clear. Explain better.*

Reply: Because fire POM and fire BC are co-emitted and assumed to be internally mixed. The burden of fire POM is about a few times higher than that of fire BC, especially in Arctic. With the removal of fire POM emission and thus fire POM in the NOFIREPOM experiment, fire BC will be impacted due to changed properties (e.g., size and hygroscopicity) of aerosol particles within which fire BC and POM are internally mixed. Our results show that the fire BC burden in the Arctic is reduced in NOFIREPOM with the mechanism worthy a detailed budget analysis. We added an explanation in the revised manuscript.

*Page 17, Line 373: it -> one*
Reply: Done.

*Page 17, Lines 375-376: global regions -> globe*
Reply: Done.

*Page 17, Lines 378-382: Could the authors provide a reference for this mechanism?*

Reply: We added the following reference:
Zhang, Z., Meyer, K., Yu, H., Platnick, S., Colarco, P., Liu, Z., and Oreopoulos, L.: Shortwave direct radiative effects of above-cloud aerosols over global oceans derived from 8 years of CALIOP and MODIS observations, Atmos. Chem. Phys., 16, 2877-2900, 10.5194/acp-16-2877-2016, 2016.

*Sect. 3.3: Can's some of the cloud changes that lead to indirect effects be a result of dynamical changes due to fire aerosols?*

Reply: Yes, the cloud changes as a result of dynamical changes due to fire aerosols is also considered as a part of aerosol induced cloud radiative effect (CRE) with the

Ghan (2013) method. Since the same sea surface temperatures (SSTs) are used in these simulations, CRE as a result of dynamical changes due to fire aerosols should be small.

*Page 19, Line 418: Please provide reference to support this statement ("Larger...").*

Reply: We added the two following references:

Ghan, S. J., Liu, X., Easter, R. C., Zaveri, R., Rasch, P. J., Yoon, J.-H., and Eaton, B.: Toward a Minimal Representation of Aerosols in Climate Models: Comparative Decomposition of Aerosol Direct, Semidirect, and Indirect Radiative Forcing, Journal of Climate, 25, 6461-6476, doi:10.1175/JCLI-D-11-00650.1, 2012.

Jiang, Y., Yang, X.-Q., and Liu, X.: Seasonality in anthropogenic aerosol effects on East Asian climate simulated with CAM5, Journal of Geophysical Research: Atmospheres, 120, 2015JD023451, 10.1002/2015JD023451, 2015.

*Page 19, Lines 420-421: What does "low-level" mean here?*

Reply: The low-level clouds mean "vertically-integrated low clouds (from surface to 750 hPa)" as defined in CESM. We revised the sentence in the manuscript to make it clear.

*Page 20, Lines 434-435: The higher OC/BC ratio does not seem like a good explanation, as it is mentioned a bit earlier that POM and BC are comparable in the NH and SH.*

Reply: we agree with the reviewer, and removed "higher fire OC/BC ratios" in the revised manuscript.

*Page 21, Line 449: I suggest adding "slightly" between "agree" and "better".*

Reply: Done.

*Page 21, Line 452: It reads as if you take values from Ghan (2013). Suggest rephrasing.*

Reply: Thanks. We rephrased the words to "estimated with *Ghan* [2013]" in the revised manuscript.

*Page 21, Lines 469-470: Even in tropical areas? Please discuss.*

Reply: We re-wrote the sentence as:

"The negative SAE over land is a result of the surface albedo change (including snow depth change) caused by fire aerosols."

*Page 22, Line 484: Instead of "The shortwave flux change in the atmosphere ", I suggest writing "The shortwave atmospheric absorption change", as it is more conventional.*

Reply: Done. Thanks for the suggestions.

*Figure 13a: Clarify to the reader why the values in Fig. 8a are somewhat different to those in Fig. 13a.*

Reply: Figure 13a shows the net shortwave flux change at TOA due to fire aerosols, which is a sum of fire aerosol DRE, CRE and SAE. The CRE (-0.70$\pm$0.05) is larger than the DRE (0.155$\pm$0.01) and SAE (0.03$\pm$0.10). Thus, the TOA solar flux change is dominant by the CRE and similar to distribution of the CRE (Figure 8a). These values are also listed and compared in Table 2.

*Page 23, Lines 505-506: There are also substantial differences with Tosca et al. (2013), especially over tropical oceans, therefore I would add "partly" before "consistent". Also the results over southern Africa are consistent with the recent findings of Hodnebrog et al. (2016), which the authors can mention.*

Reply: Thanks for the suggestions. We added the word "partly" and also the sentence that "The precipitation reduction in southern Africa is consistent with the recent findings of *Hodnebrog et al.* [2016]" in the revised manuscript.

*Page 23, Line 511: After this line, I suggest that you add a statement clearly stating that these results do not represent the complete impact of fire emitted aerosols on temperature and (especially) precipitation, since the climate system has not been allowed to fully respond (SSTs are fixed).*

Reply: Thanks for the suggestion. We added a statement in the revised manuscript: "We note that the temperature and (especially) precipitation changes reported here do not represent the complete impact of fire aerosols, since the SSTs are fixed in our simulations. Fully-coupled atmosphere and ocean model will be used to further investigate the impact of fire aerosols."

*Page 24, Line 519: effect -> effective*
Reply: Done.

*Page 26, Lines 575-579: Again, I suggest reminding the reader that these do not represent the full climate responses, given the atmosphere-only nature of the experiments.*
Reply: following the reviewer's comment, we added a statement here in the revised manuscript "These results are based on the simulations with fixed SSTs and may not represent the full climate responses."

*Page 27, Lines 596-597: Is the difference in emitted POM between the two studies equivalent (in size) to the difference in the CRE?*

Reply: The CRE is strongest over southern Africa, South America and the Arctic. The emission scaling factors used in *Ward et al.* [2012] for these three regions are 3, 2 and 3, respectively. The CRE of their study is about 2.4 times of our study (-1.64 versus -0.70 W m$^{-2}$). So the difference in CRE between the two studies is approximately equivalent (in size) to the emission difference.

---

## Author Comment (AC2) · 7 Aug 2016

We thank the reviewer for his/her careful reviews and helpful comments. The manuscript has been revised accordingly and our point-by-point responses are provided below. (Reviewer's comments are in italic and the responses in standard font).

**Reviewer #1**

*This paper examines the global and regional radiative forcings by black carbon and organic carbon aerosols from open fires. The authors use the NCAR Community Atmosphere Model version 5.3 (CAM5) with the four-mode version of the modal aerosol module (MAM4) and employ two methods to calculate forcing. In one method, they follow Ghan et al. (2013), which may produce a more robust estimate of forcing. In the second method, they follow a more traditional approach. The authors find that top-of-atmosphere (TOA) forcing from aerosol-cloud interactions dominates the total global forcing (-0.70 W $m^{-2}$). When aerosol-radiation interactions and aerosol effects on snow are also considered, the global annual mean forcing from open fire aerosols is -0.55 W $m^{-2}$. The authors also estimate the climate impacts of fire aerosols.*

*The paper leads to no startling new conclusions, but may provide a more accurate estimate of the global and regional climate impacts of aerosols from open fires. The paper should be revised in response to the major criticisms and resubmitted.*

Reply: We thank the reviewer for helpful comments. The manuscript is revised following the comments and criticisms from the reviewer.

*Major criticisms.*

*1. The paper needs to make more clear what is new in the results, or why this approach represents a substantial improvement over previous results. Central to this paper should be the answer to this question: Why does this research give us greater confidence in our knowledge of the effects of fire aerosols on climate?*

*In Lines 147-150, the text lists a few improvements, but supplies little elaboration. The improvements are: (a) higher spatial resolution, (b) use of the latest CAM5 model with updated MAM4, (c) calculation of daily instead of monthly fire emissions, and (d) use of an alternative methodology to calculate radiative forcings of aerosols (Ghan 2013). It's not clear why the relatively small increase in spatial resolution would lead to better results, or why calculation of daily instead of monthly fires matters. Almost no information on the updates in MAM4 is given or what difference they make for forcing calculations. A detailed explanation of the benefits of the Ghan (2013) method over other methods is absent.*

→Reply: We thank the reviewer for the comments. We now make it more clear what is new in our results, and why our approach represents a substantial improvement over previous results in the revised manuscript.

Specially, following the reviewer's comment, we elaborate more on the improvements of our approach and model configuration in the revised manuscript:

(a) *higher spatial resolution*. A model resolution change from 2 degree (used in previous studies) to 1 degree (in this study) represents a resolution increase by 4 times. A higher resolution allows more efficient transport of aerosols from the sources to remote regions due to reduced wet scavenging of aerosols as a result of less frequent collocation between aerosols and clouds at higher resolutions (Ma et al., 2013; 2014). Model resolution has also been shown to be important for aerosol radiative forcing due to aerosol-cloud interactions (Ma et al., 2015).

(b) *use of the latest CAM5 model with updated MAM4*. Compared to the 3-mode version of MAM (MAM3) used in previous studies, MAM4 includes a primary carbon mode to explicitly treat the microphysical ageing of primary carbonaceous aerosols (POM/BC) in the atmosphere. Primary carbonaceous aerosols are emitted in the primary carbon mode and transferred to the accumulation mode due to aerosol condensation and coagulation. Because of a lack of primary carbon mode, MAM3 assumes that primary carbonaceous aerosols are emitted in the accumulation mode and thus instantaneously mixed with other soluble aerosol species (e.g., sulfate), subject to wet scavenging in the accumulation mode. As a result, MAM4 has higher BC and POM burdens over MAM3 in the remote regions by ~30%.

(c) *calculation of daily instead of monthly fire emissions*. Using daily emissions will allow the model to consider the effect of fast changes in fire emission flux on the local atmospheric conditions. It is expected that using the monthly mean emission flux the model can't consider the effect of the extremely strong fires, thus it might underestimate the fire forcing for such cases. Considering that the aerosol effect is often non-linear, using higher temporal resolution emission data will make a difference, at least for the effect on daily extremes.

(d) *use of an alternative methodology to calculate radiative forcings of aerosols (Ghan 2013)*. Ghan (2013) provides a more accurate method to calculate the radiative forcing of aerosols. Central to this method is that the radiative forcing due to aerosol-radiation interactions must be calculated in the presence of clouds (i.e., under all-sky condition, $\Delta(F - F_{clean})$), and the radiative forcing due to aerosol-cloud interactions be calculated under the condition of no aerosol effects on radiation (i.e., $\Delta(F_{clean} - F_{clean,clear})$). $F_{clean}$ is calculated from the diagnostic radiation call with aerosol scattering and absorption neglected, and $F_{clean,clear}$ from the diagnostic radiation call with both aerosol and cloud scattering and absorption neglected. With the radiative forcing decomposition of this method, the impact of aerosols on surface albedo is also quantified (i.e., $\Delta F_{clean,clear}$).

In addition to the above improvements in model configuration and approach of

calculating radiative forcings, we validate the model performance through a comparison of our modeled AOD and SSA with the AERONET data; modeled radiative forcing due to aerosol-radiation interactions compared with satellite-derived estimations, and modeled BC-in-snow concentrations with observations in Northern China and the Arctic. These model improvements and evaluations give us greater confidence in our knowledge of the effects of fire aerosols on climate.

Some notable key findings from this study are highlighted in the conclusion section:
a) Fire aerosol radiative effect due to ARI in the Arctic regions ($0.428\pm0.028$ W m$^{-2}$) is larger than that in the tropical regions ($0.172\pm0.017$ W m$^{-2}$), although the fire aerosol burden is largest in the tropics, which results from the larger amount of low clouds in the Arctic.
b) The large cloud liquid water path over land areas and low solar zenith angle of the Arctic favor the strong fire aerosol radiative effect due to ACI (up to -15 W m$^{-2}$) during the Arctic summer.
c) The global annual mean surface albedo effect (SAE) of fire aerosols over land areas ($0.03\pm0.10$ W m$^{-2}$) is relatively small and insignificant.
d) The fire aerosols reduce the global mean surface air temperature ($T_s$) by $0.03\pm0.03$ K and precipitation by $0.01\pm0.002$ mm day$^{-1}$. Significant reductions of precipitation in southern Africa and NH high-latitudes are noticed.

*2. The paper uses outdated terms to describe radiative forcing by aerosol, and does not adequately describe what adjustments to the model meteorology have been allowed in the forcing calculations. Following IPCC AR5, the authors should use the terms aerosol-radiation interactions (ARI), aerosol-cloud interactions (ACI), and forcings due to surface albedo changes (Boucher et al., 2014; Myhre et al., 2014). ACI in the IPCC framework includes the effects of aerosols on cloud droplet number, cloud lifetime and takes into account the "semi-direct effect" of absorbing aerosols. The ACI category of forcings is useful as it makes it unnecessary to distinguish between the sometimes competing effects of aerosols on clouds.*

→Reply: Thank for the suggestion. Following the reviewer's comment, we now use the terminology of the radiative forcings by aerosol from IPCC AR5 in the revised manuscript. In our results, the cloud radiative effect (CRE), i.e., radiative effect due to aerosol-cloud interactions (ACI) includes the effects of aerosols on cloud droplet number and cloud lifetime through acting as CCN, and the semi-direct effect of absorbing aerosols.
All the atmospheric variables (including temperature, precipitation, and circulation) are allowed to adjust. However, with sea surface temperatures (SST) and sea ice are prescribed in the simulations, only the rapid adjustments are taken into account. We have made it clearer in the revised manuscript.

*The authors should further state whether they calculated radiative forcings (RF) or effective radiative forcings (ERF), which take into account the rapid adjustments to a*

*range of meteorological variables. If these are ERFs (and they seem to be), the authors need to make clear what meteorological variables they allowed to adjust. The authors should emphasize in the abstract and conclusions that the forcings they report are relative to the case of no fires, and not to conditions in 1750s.*

→Reply: Yes, with the method of Ghan (2013), the effective radiative forcings (ERF) are calculated in this study. All the atmospheric variables (including temperature, precipitation, and circulation) are allowed to adjust. However, with sea surface temperatures (SST) and sea ice are prescribed in the simulations, only the rapid adjustments are taken into account. We also emphasize in the abstract and conclusions that the radiative effects we report are relative to the case of no fires. We now use the term "radiative effect" instead of "radiative forcing" of fire aerosols throughout the text.

*3. It's not clear why the paper does not consider the effects of fire aerosols on sea ice albedo. Is this not an important forcing term? Also the authors neglect the issue of brown carbon, which has recently been suggested as a main component of primary organic matter (POM) in fire plumes (Feng et al., 2013). MAM4 may not be capable of simulating brown carbon, and this should be acknowledged.*

→Reply: In our simulations with the stand-alone CAM5, sea surface temperatures and sea ice are prescribed, and thus the effects of fire aerosols on sea ice albedo are not considered. The effects of fire aerosols on sea surface temperatures and sea ice albedo will be presented in our future study using a slab ocean model coupled with CAM5.

The effects of POM as brown carbon are not considered in MAM4, and we acknowledge this in the revised manuscript.

*4. The authors report a large number of changes in global mean variables without giving uncertainty ranges or stating which changes are statistically significant. Given that many of the variables have been calculated using an ensemble of simulations, uncertainties should be easy to calculate.*

→Reply: Following the reviewer's comment, we added the uncertainty ranges (±1σ uncertainty) for changes in global mean variables in the revised manuscript.

***Other criticisms.***

*Title: Given the distribution of fires in Figure 2, it looks like the authors include agricultural fires in their analysis, and so the term "wildfire" should be changed to "open fires."*

→Reply: Yes, the agricultural fires are included. We changed the term "wildfire" to

"open fires".

*Abstract. The abstract should state the time period under investigation. Also large regional forcings should be quantified, as they could have importance for regional climate.*

→Reply: We added the time period (2003-2011) in the abstract. Also the following sentence is added in the abstract for large regional forcings: "REs due to fire ARI and ACI in the Arctic (0.43±0.03 and -0.82±0.09 W m$^{-2}$, respectively) are stronger than those in the tropics (0.17±0.02 and -0.70±0.05 W m$^{-2}$, respectively), although the fire aerosol burden is higher in the tropics."

*Introduction. The introduction is too long. The first paragraph should make clear exactly what problem is being considered, and it should succinctly explain why this investigation represents a major improvement over past research. Throughout the introduction, many old references brought up – e.g., Chuang et al. (2002) or IPCC AR4. The authors should condense the introduction and focus on Chapters 7 and 8 in AR5 and subsequent papers – e.g., Myhre and Samset (2015), Chakrabarty et al. (2014), and many others. Missing from the introduction is a discussion of the radiative effects of organic vs black carbon.*

→Reply: Thanks for the suggestions. Following the reviewer's comment, we made it clear in the first paragraph what problem is being considered in this study by adding the sentence: "An qualification of rediative forcing of fire aerosols is the first step to reduce these uncertainties [*Ward et al.,* 2012]".
We added the explanation why this investigation represents a major improvement over past research (see our response to the reviewer's major criticism #1).
We condensed the introduction and focused on Chapters 7 and 8 in AR5 and subsequent papers. We removed the old references, e.g., Chuang et al. (2002) or IPCC AR4 in the revised manuscript.

The following dicussion of BC and POM's radiative effects are added: "Although there are many studies quantifying the RE of fire aerosols, a further investigation is still needed, as current estimations of the RE of fire aerosols from climate models are still associated with large uncertainties [*Myhre and Samset*, 2015; *Chakrabarty et al.*, 2014], and the REs of fire POM versus BC are even less clear."

*Line 174. The authors state that MAM4 "significantly increases (and improves) the BC concentrations in the Arctic…." Why does inclusion of the primary carbon mode in MAM4 improve the treatment of microphysical aging of BC? How did the authors decide that inclusion of this mode "significantly" improves the BC simulation? By what measure? Elsewhere the authors state that MAM4 "realistically represents the external/internal mixing of BC" (Line 578). But no detail is given about these improvements.*

→Reply: In the 3-mode version of MAM (MAM3), due to a lack of primary carbon mode, BC is emitted directly into the accumulation mode, and thus is instantaneously mixed with other soluble aerosol species (e.g., sulfate), subject to wet removal by clouds and precipitation. MAM4 includes an additional primary carbon mode on top of MAM3. BC is emitted in this primary carbon mode, and is gradually transferred to the accumulation mode due to the microphysical aging (condensation and coagulation). Aerosol in the primary carbon mode is less hygroscopic than that in the accumulation mode, and thus is less susceptible to the wet scavenging by clouds. Therefore, BC concentration from MAM4 is increased, especially in the Arctic, which improves the agreement with observations. The details of MAM4 and comparison with MAM3 are given in Liu et al. (2016). Please see also our reply to the major criticism #1 for the description of BC representation in MAM4 versus in MAM3.

We added the following details in the introduction of the revised manuscript: "MAM4 includes an additional primary carbon mode on the top of MAM3 to explicitly treat the microphysical ageing of primary carbonaceous aerosols (POM and BC) in the atmosphere. POM and BC in MAM4 are emitted in the primary carbon mode instead of the accumulation mode as in MAM3. Thus MAM4 increases the BC and POM concentrations over MAM3 due to reduced wet scavenging of POM and BC in the primary carbon mode with a lower hygroscopicity than that in the accumulation mode."

*Section 2.3. See major criticism #2 above. Please rewrite using IPCC AR5 convention for describing forcings.*

→Reply: Done. See our reply above to the major criticism #2.

*Results. The results section rambles. The authors should decide which are the key results and provide more detailed explanations of the mechanisms driving these results. Also, the statistical significance of results should be given, where possible. Since the authors performed an ensemble of simulations, many results can be reported with one standard deviation uncertainty. For example, what is the uncertainty of the forcings calculated following Ghan 2013? Is the -0.03°C temperature effect of fire aerosols statistically significant?*

→Reply: Thanks for the suggestions. We revised the results section and emphasized the key results. Please see our response above to the major criticism #1 for the key results. We have provided more detailed explanations of the mechanisms driving these results.

Following the reviewer's comment, we added the statistical significance of results with one standard deviation uncertainty. This is done for the uncertainty of the forcing calculated following Ghan (2013) as well as the temperature and precipitation

changes due to fire aerosols.

*Finally, the forcings calculated for specific regions should be compared to recently published estimates – e.g. Brieder et al. (2014) for the Arctic and Sena and Artaxo (2015) for South America.*

→Reply: Thanks for the suggestion. We tried to compare our forcings with those estimated from Brieder et al. (2014) for the Arctic. However, we found that this study reported the distribution, aerosol optical depth, and absorption of Arctic aerosol components and source contributions calculated using the GEOS-Chem model, and did not present the forcing estimates.

Following the reviewer's comment, we added the following comparison of our forcing estimates with those from Sena and Artaxo (2015) for South America in the revised manuscript: "The fire aerosol RE due to ARI over South America for the period of 2000 to 2009 is estimated with the TOA shortwave flux from CERES (Clouds and Earth's Radiant Energy System) and AOD from MODIS by Sena and Artaxo (2015). The clear-sky RE during the fire season (August to September) is estimated to be -5.2 W m$^{-2}$, which is larger than our result (-2.1 W m$^{-2}$). This is consistent with the underestimation of our modeled AOD in South America when compared to the AERONET data (Figure 3)."

*Line 241. Here and elsewhere. It is not clear whether the fires examined in this study include agricultural fires such as those in Equatorial Asia and South America.*

→Reply: Yes, the agricultural fire is included. We made it clear in the revised manuscript.

*Lines 276-on. The text should state whether the modeled AOD includes aerosol from all sources, not just fires.*

→Reply: The modeled AOD includes aerosol from all sources. We made it clear in the revised manuscript.

*Line 311. The text states, "Although MAM4 increases the column burdens of POM and BC by up to 40% in many remote regions compared to MAM3…." Why does this large increase occur?*

→Reply: see our response above for the explanation of MAM4 and MAM3 simulated BC differences.

*Line 338. Text should be more clear about how clouds amplify the forcing of BC.*

→Reply: We added the following explanation in the revised manuscript: "When BC resides above clouds, its absorption of solar radiation is significantly enhanced due to the reflection of solar radiation by clouds [*Abel et al.*, 2005; *Zhang et al.*, 2015]".

*Line 343. Why is the forcing estimated from Terra different from that of Aqua?*

→Reply: First of all, we notice that we had a wrong subtitle in Figure 7b and Figure 7c. Figure 7b should be for Aqua/MODIS, and Figure 7c should be for Terra/MODIS. The figure caption is accurate in the text.

Over southeastern Atlantic, smoke aerosols usually reside above the stratocumulus clouds. Therefore, the direct radiative forcing strongly depends on the underlying cloud fraction. If the cloud fraction is higher, for the same amount of smoke aerosols at exact the same altitude, smoke aerosols can exert stronger direct radiative forcing. Since stratocumulus clouds over this region exist the diurnal cycle, the forcing estimated from Terra (morning time, with larger amount of clouds) is different from the one estimated from Aqua (afternoon time, with smaller amount of cloud). For more detail, we recommend the reviewer to check Figure 3 in the reference:
Min M., and Zhang Z. (2014), On the influence of cloud fraction diurnal cycle and sub-grid cloud optical thickness variability on all-sky direct aerosol radiative forcing, J. Quant. Spectros. Radiat. Transfer, doi:10.1016/j.jqsrt.2014.03.014.

*Line 346. There is no mention here or elsewhere about the effect of solar zenith angle on radiative forcing at high latitudes, particularly the Arctic.*

→Reply: We agree with the reviewer that the cloud radiative forcing due to fire aerosols at high latitudes can be affected by the solar zenith angle (Shupe et al., 2004). In the boreal summer, the lower solar zenith angle favors the larger DRE in the Arctic. We added this effect in the revised manuscript.

*Line 349. Here and elsewhere, the authors should take care with the terms "summer" and "autumn" when referring to the Southern Hemisphere.*

→Reply: Thanks. We made it clearer in the revised manuscript. All terms were changed to "boreal summer" or "boreal autumn".

*Line 354. "noises" Please fix English.*
→Reply: Thanks. We changed to "…, and there is much less noise".

*Line 364. The text states: It is not clear why removal of POM in the simulation affects BC concentrations. If indeed this is what happens, then the Ghan method for calculating forcing should not be used for individual fire components.*

→Reply: Because fire POM and fire BC are co-emitted and assumed to be internally

mixed. The burden of fire POM is about a few times larger than that of fire BC, especially in Arctic. With the removal of fire POM emission and thus fire POM in the NOFIREPOM experiment, fire BC will be impacted due to changed properties (e.g., size) of aerosol particles within which co-emitted fire BC is internally mixed with fire POM. Our results show that the fire BC burden in the Arctic is reduced in NOFIREPOM with the exact mechanism warranty of a detailed budget analysis. We added an explanation in the revised manuscript.

We would like to keep the Ghan method for calculating the radiative effects of individual fire components (POM and BC). The reason is that the Ghan method only introduces the relatively large bias for fire POM radiative effect (due to aerosol-radiation interactions), and the bias for fire BC radiative effect is small (comparing the Ghan and the BBFFBF methods). By using the two different methods we will be able to examine the uncertainty range of radiative effects of individual fire components. Also the Ghan method allows us to calculate the radiative effects of individual fire components due to aerosol-cloud interactions.

*Line 379. See above comment.*
*Cloud radiative forcing section. Please see major criticism #2. Also, this section should provide discussion of why the forcing due to ACI is stronger in some regions compared to others.*

→Reply: Please see our responses to the major criticism #2 above.

We added the following discussion of why the forcing due to ACI is stronger in some regions compared to others in the revision: "The different spatial distributions of fire aerosol radiative effect (RE) due to ACI in the NH high latitudes and in the tropics result from the difference in cloud distributions between the two regions. During the fire season the cloud LWP over the land areas in the NH high latitudes is three times larger than that over the ocean areas in the tropics. Larger cloud LWP favors the stronger RE due to ACI, because the larger LWP associated with the warm cloud and rain processes favors the aerosol effect on slowing down the autoconversion of cloud water to rain [*Ghan et al.*, 2012; *Jiang et al.*, 2015]. Meanwhile, in the Arctic, the low solar zenith angle in summer favors the large fire aerosol RE due to ACI."

*Line 411. The text should state why larger cloud liquid water path leads to stronger forcing due to ACI.*

→Reply: We added the following explanation: "Larger cloud LWP favors the stronger RE due to ACI, because the larger LWP associated with the warm cloud and rain processes favors the aerosol effect on slowing down the autoconversion of cloud water to rain [*Ghan et al.*, 2012; *Jiang et al.*, 2015]."

*Section on surface snow albedo forcing. Why are forcings due to BC deposition on sea ice not considered? The section seems misnamed, since forcings on all light colored*

*surfaces are seen in Figure 12.*

→Reply: In our simulation, the sea surface temperature and sea ice is prescribed, and thus the radiative effect due to fire BC deposition on sea ice is not estimated.

We rename the title of the section to "Surface albedo effect". The surface albedo change not only results from the radiative effect of fire BC deposition on snow albedo, but also from atmospheric feedbacks (e.g., snow depth change and snow melting) due to fire aerosols.

*The forcings on surface albedo calculated with the Ghan 2013 method look suspiciously high over low latitudes (Figure 12). The authors should comment on these high values – e.g., +0.5 $Wm^{-2}$ over parts of the U.S. south. Are these results comparable to those from SNICAR?*

→Reply: The SAE of fire aerosols is also noticed over low latitudes, which includes the surface albedo changes from atmospheric feedbacks (e.g., snow depth change and snow melting) [*Ghan,* 2013]. These high values over low latitudes are not evident in those from SNICAR, which are diagnosed in the standard model simulation and don't include atmospheric feedbacks. We added a comment on these high values in the revised manuscript.

*Figure 12b reveals no significant differences in forcings for the fire vs no-fire cases over the Arctic or north China. The authors should acknowledge this. Given the results from SNICAR, it seems that the only region that might show a significant impact of fire aerosols on surface albedo is Greenland and the very northern reaches of Canada.*

→Reply: The annual mean fire BC forcing in the Arctic and North China (~ 0.01 W $m^{-2}$) is much smaller than that in Greenland and the very northern reaches of Canada. It is because the snow-covered time of Arctic and North China is shorter. The forcing in these two regions (Greenland and the very northern reaches) can reach up to 0.5 W $m^{-2}$. We acknowledged this in the revised manuscript.

*Line 458. It sounds like snow melting is one of the rapid meteorological adjustments allowed to occur in the forcing calculation. Is this correct?*

→Reply: Yes, the snow melting is allowed when calculating the surface albedo effect of fire aerosols.

*Section on the fire aerosol effects on shortwave radiation, global temperature and precipitation. Here the statistically significance and the uncertainties of global results should be stated. If the global mean changes of some variables are not statistically significant, then that should be made clear.*

→Reply: We added the significant information (e.g., one-standard deviations) in the text and in Table 2. The global mean changes not statistically significant are acknowledged in the revised manuscript.

*Discussion section. Again the authors should stress the key points and put them in context of other new studies besides just Ward 2012 and Tosca 2013. What exactly is new in this study? Limitations and uncertainties of the study should be discussed – i.e., what are the shortcomings of the approach used here?*

→Reply: We have included a discussion of the key points of this study as summarized as follows:

a)  Fire aerosol RE due to ARI in the Arctic regions ($0.428 \pm 0.028$ W m$^{-2}$) is larger than that in the tropical regions ($0.172 \pm 0.017$ W m$^{-2}$), although the fire aerosol burden is higher in the tropics. This results from the larger low cloud amount in the Arctic;

b)  The large cloud liquid water path over land areas, and low solar zenith angle of the Arctic favor the strong fire aerosol RE due to ACI (up to -15 W m$^{-2}$) during the Arctic summer;

c)  The global annual mean surface albedo effect (SAE) over land areas ($0.03 \pm 0.10$ W m$^{-2}$) is relatively small and insignificant;

d)  The fire aerosols reduce the global mean surface air temperature ($T_s$) by $0.03 \pm 0.03$ K and precipitation by $0.01 \pm 0.002$ mm day$^{-1}$. Especially, significant reductions of precipitation in southern Africa and in the NH high-latitudes are noticed.

Following the reviewer's comment, we added a discussion of limitations and uncertainties of this study:

1)  The RE estimate of co-emitted fire POM with the Ghan (2013) approach is not accurate due to the assumption of internal mixing of individual fire components (POM and BC);

2)  There is large noise associated with the surface albedo effects of fire aerosols with the Ghan (2013) approach due to the snow melting and atmospheric feedbacks;

3)  There are uncertainties with the model simulation and configuration. For example, the model still underestimates observed AODs (mostly within a factor of 2) at the sites predominantly influenced by biomass burning aerosols during the fire season. It implies that the fire aerosol radiative effects can be stronger than those estimated in this study. In our simulation, the sea surface temperature and sea ice is prescribed, and the fire BC effects on sea ice is not considered. The brown carbon component of POM [Feng et al., 2013] is not considered in our current simulations, which may result in an underestimation of atmospheric absorption of fire aerosols."

***Tables and Figures.***

*There are too many Figures. Decide what is important and put rest in a supplement.*
→Reply: We moved the original Figure 2 (POM and BC burdens from different sources) and Figure 7 (fire aerosol radiative effect due to ARI at four seasons) to the supplement.

*Captions should be stand-alone so that the browsing reader can understand what is being shown. Unusual acronyms should be explained.*
→Reply: We added the standing-alone captions of all figure and tables at the end of the manuscript. We removed some unusual acronyms and added explanations for the others in the revised manuscript.

*Units in Table 2 should be within the table, not in the caption.*
→Reply: Done.

*Uncertainty ranges should be included in Table 2, and significant changes shown in boldface.*
→Reply: We revised Table 2 to include the uncertainty ranges and those significant changes are shown in boldface.

*Text on all legends should be large enough to read. The latitude and longitude labels on the global maps can be eliminated for a cleaner, less cluttered appearance.*
→Reply: We enlarged the text on legends of the figures. The duplicated latitude and longitude labels on the global maps were eliminated.

*Global mean values should be reported to 2-3 significant digits.*
→Reply: The global mean values were now reported to 3 significant digits.

*Figures 4 and 5 should include error bars.*
→Reply: Done.

*Figure 7. What does white space represent?*
→Reply: White space represents the missing values. As we mentioned in the figure caption, the radiative effect is estimated for above-cloud aerosols only. During the fire season, cloud fractions over the land, especially below 10°S, are extremely low, and close to 0. No above-cloud smoke aerosols were detected by satellites over these regions; therefore, no radiative effect due to above-cloud aerosols is estimated.

*Figure 14. Replace acronyms above the panels with standard English terms.*
→Reply: Done.

---

## Referee Report (RR1)

Review of "Impacts of global open fire aerosols on direct radiative, cloud and surface-albedo effects simulated with CAM5," by Jiang et al.

In this paper, Jiang et al. use a global climate model and two approaches to estimate the radiative forcing and climate impacts of aerosols from fires. They consider both black carbon (BC) and primary organic matter (POM) and all three categories of radiative effects – from aerosol-radiation interactions (ARI), aerosol-cloud interactions (ACI), and the effects of BC on snow. The approach used by the authors represents an improvement over past model studies since they use an updated aerosol module, finer spatial resolution, and two approaches to calculate radiative effects. The authors report an annual mean ARI radiative effect from fire aerosols of $0.16\pm0.01$ W m$^{-2}$, and an ACI radiative effect of -0.70$\pm$0.05 W m$^{-2}$. They find a decrease in global mean surface temperatures by -0.03 K, with local cooling as large as -1 K. Precipitation also decreases in some regions due to a more stable boundary layer and reduced convection.

The authors have carefully responded to the main criticisms from the first round of reviews, and I have only minor comments. For example, the paper now clearly states how this work builds on prior model studies, and it describes the mechanisms that drive the spatial and temporal patterns of radiative effects and climate response.

Minor criticisms.
Line 280-282. Why is the OC to BC ratio in emissions of forest fires almost 3 times higher than that from other kinds of fires (grassland, savannah, and deforestation)?

Lines 428-431. The authors note that the sum of ARI radiative effects from individual components (BC and POM) is greater than the radiative effects due to all aerosols. They then state that this is evidence of nonlinear interactions among aerosol components. The reader would appreciate more details on these nonlinear interactions, perhaps an example.

Lines 450-452. Again some explanation or examples of nonlinear interactions affecting ACI radiative effects would be helpful.

---

## Author Response (AR2)

We thank the reviewer for his/her careful reviews and helpful comments. The manuscript has been revised accordingly and our point-by-point responses are provided below. (Reviewer's comments are in italic and the responses in standard font).

*Minor criticisms.*
*Line 280-282. Why is the OC to BC ratio in emissions of forest fires almost 3 times higher than that from other kinds of fires (grassland, savannah, and deforestation)?*

→Reply: The OC to BC ratio in emissions strongly depends on the burning phases (smoldering versus flaming phases). For forest fires, most of the emissions come from the smoldering phase of burning, which has a higher OC to BC ratio. For other kinds of fires (grassland, savannah, and deforestation), the emissions come mainly from the flaming phase of burning, which yields a lower OC to BC ratio.
We have added a note in the revised manuscript.

*Lines 428-431. The authors note that the sum of ARI radiative effects from individual components (BC and POM) is greater than the radiative effects due to all aerosols. They then state that this is evidence of nonlinear interactions among aerosol components. The reader would appreciate more details on these nonlinear interactions, perhaps an example.*

→Reply: In the MAM4 aerosol module, individual aerosol components (e.g., fire BC and fire POM) are internally mixed within aerosol modes, thus different aerosol components can influence each other in the ARI radiative effects. An example of nonlinear interactions among aerosol components includes: fire POM and water on fire BC particles enhance solar absorption by the fire BC. This will make the sum of radiative effects from fire POM and BC greater than that due to all fire aerosols.
We have added a note in the revised manuscript.

*Lines 450-452. Again some explanation or examples of nonlinear interactions affecting ACI radiative effects would be helpful.*

[revised manuscript text omitted]